# SLIT2/ROBO1-signaling inhibits macropinocytosis by opposing cortical cytoskeletal remodeling

Vikrant K. Bhosle [1], Tapas Mukherjee [2], Yi-Wei Huang [1], Sajedabanu Patel [1], Bo Wen (Frank) Pang[1,3,14], Guang-Ying Liu[1], Michael Glogauer[4,5,6], Jane Y. Wu[7,8,9], Dana J. Philpott[2], Sergio Grinstein [1,10,11] & Lisa A. Robinson [1,3,12,13 ✉]

Macropinocytosis is essential for myeloid cells to survey their environment and for growth of RAS-transformed cancer cells. Several growth factors and inflammatory stimuli are known to induce macropinocytosis, but its endogenous inhibitors have remained elusive. Stimulation of Roundabout receptors by Slit ligands inhibits directional migration of many cell types, including immune cells and cancer cells. We report that SLIT2 inhibits macropinocytosis in vitro and in vivo by inducing cytoskeletal changes in macrophages. In mice, SLIT2 attenuates the uptake of muramyl dipeptide, thereby preventing NOD2-dependent activation of NF-κB and consequent secretion of pro-inflammatory chemokine, CXCL1. Conversely, blocking the action of endogenous SLIT2 enhances CXCL1 secretion. SLIT2 also inhibits macropinocytosis in RAS-transformed cancer cells, thereby decreasing their survival in nutrient-deficient conditions which resemble tumor microenvironment. Our results identify SLIT2 as a physiological inhibitor of macropinocytosis and challenge the conventional notion that signals that enhance macropinocytosis negatively regulate cell migration, and vice versa.

[1] Program in Cell Biology, The Hospital for Sick Children, Peter Gilgan Centre for Research and Learning, 686 Bay Street, Toronto, ON M5G 0A4, Canada. [2] Department of Immunology, University of Toronto, Medical Sciences Building, 1 King's College Circle, Toronto, ON M5S 1A8, Canada. [3] Institute of Medical Science, University of Toronto, Medical Sciences Building, 1 King's College Circle, Toronto, ON M5S 1A8, Canada. [4] Faculty of Dentistry, University of Toronto, 101 Elm Street, Toronto, ON M5G 2L3, Canada. [5] Department of Dental Oncology and Maxillofacial Prosthetics, University Health Network, Princess Margaret Cancer Centre, 610 University Avenue, Toronto, ON M5G 2C1, Canada. [6] Centre for Advanced Dental Research and Care, Mount Sinai Hospital, 600 University Avenue, Toronto, ON M5G 1X5, Canada. [7] Department of Neurology, Northwestern University Feinberg School of Medicine, Chicago, IL 60611, USA. [8] Center for Genetic Medicine, Northwestern University Feinberg School of Medicine, Chicago, IL 60611, USA. [9] Lurie Cancer Center, Northwestern University Feinberg School of Medicine, Chicago, IL 60611, USA. [10] Department of Biochemistry, University of Toronto, Medical Sciences Building, 1 King's College Circle, Toronto, ON M5S 1A8, Canada. [11] Keenan Research Centre of the Li Ka Shing Knowledge Institute, St. Michael's Hospital, 290 Victoria Street, Toronto, ON M5C 1N8, Canada. [12] Division of Nephrology, The Hospital for Sick Children, 555 University Avenue, Toronto, ON M5G 1X8, Canada. [13] Department of Paediatrics, Faculty of Medicine, University of Toronto, 555 University Avenue, Toronto, ON M5G 1X8, Canada. [14]Present address: BenchSci, Suite 201, 559 College Street, Toronto, ON M6G 1A9, Canada. ✉email: lisa.robinson@sickkids.ca

Macrophages, which are specialized cells of the innate immune system, perform diverse functions, including host defense against pathogens[1]. Pro-inflammatory macrophages use various tools to combat invading microorganisms, including secretion of inflammatory mediators, phagocytosis of the pathogen, and production of reactive oxygen species. These cellular processes are carefully regulated by dynamic changes in the macrophage cytoskeleton[2]. During immune surveillance, macrophages, and immature dendritic cells (iDCs) constitutively sample their extracellular surroundings via macropinocytosis, a phenomenon brought about by active plasma membrane ruffling induced by remodeling of the cortical cytoskeleton[3,4]. These cells also internalize bacterial pathogen-associated molecular patterns (PAMPs), such as muramyl dipeptide (MDP), via macropinocytosis, thereby initiating an immune response[5]. Intracellular pathogens, including some viruses, exploit macropinocytosis to facilitate their entry into the host cells[6,7]. Several growth factors, including colony stimulating factor 1 (CSF1), secreted by the host, as well as some pathogen-derived proteins such as IpaC from *Shigella* are known to induce macropinocytosis[3,8,9]. It is estimated that macrophages can consume liquid equal to their cell volume every hour[10]. This would leave them particularly vulnerable to the entry of soluble PAMPs and viruses, via macropinocytosis, under specific conditions.

Macropinocytosis is a near-universal feature of cancers driven by mutations in the RAS family of oncogenes[11–13]. RAS-transformed tumors account for approximately one third of all malignancies in humans[14]. Among the RAS family members, KRAS mutations are the most common in cancer, especially in adenocarcinomas such as pancreatic adenocarcinoma (PDAC)[15], and they are associated with poorer response to the conventional therapies and worse prognosis[16]. KRAS-transformed cancer cells use macropinocytosis to internalize large proteins, such as albumin, from their extracellular environment, to subsequently break them down into amino acids, which enter the cell metabolism to promote cancer cell survival[11]. This might be particularly relevant in the tumor core, which contains low concentrations of amino acids, including glutamine (Glut)[17]. Inhibition of macropinocytosis could be of therapeutic benefit in this context[11]. However, no physiological factor that can inhibit macropinocytosis has been identified.

Spatiotemporal regulation of activity of Rho-family of small GTPases is essential to bring about localized and reversible changes in the cellular actin cytoskeleton[18]. Accordingly, multiple steps during macropinocytosis are regulated by small Rho GTPases, Rac1 and Cdc42[3,19]. The conserved Slit family of neuronal guidance cues, together with their transmembrane Roundabout (Robo) receptors, act as repellents during development of the central nervous system by regulating actin cytoskeletal rearrangements in migrating neurons and projecting axons[20]. Slit proteins (SLIT1-3) undergo proteolytic cleavage in vivo, with N-terminal fragments binding to Robo receptors to induce signaling[21]. A growing number of recent studies suggest that Slit-Robo signaling also has potent, localized, tissue-specific effects outside the nervous system[22–25]. Of the four mammalian Robo receptors (ROBO1-4), ROBO3 does not bind to Slit proteins[26,27], and ROBO4 is exclusively expressed in endothelial cells[28]. Among Slit proteins, SLIT1 is predominantly expressed in the nervous system, while SLIT2 and to a lesser extent SLIT3 are also found in peripheral tissues[24,29]. We and others have shown that SLIT2, together with ROBO1, inactivates Rac1/2 to regulate actin networks in immune cells, and as a result, inhibits the directed migration of neutrophils[29–31], monocytes[32], dendritic cells[33], and T lymphocytes[34] toward chemotactic stimuli in vitro and in vivo[31–33,35]. Although primary murine macrophages were recently shown to express ROBO1 and ROBO3[24], the actions of SLIT2 on macrophage functions have been largely unexplored.

The role played by Slit-Robo signaling in tumor progression vs. suppression also remains a topic of active investigation and appears to be context dependent[36,37]. Nonetheless, *SLIT2* has been shown to be silenced in several invasive tumors and in cancer cell lines, including PDAC cells, and conversely, high levels of *SLIT2* mRNA are associated with suppressed tumor growth in vivo[38,39]. Although SLIT2-induced tumor suppression has been presumed to be due inhibition of cancer cell migration, its effects on cancer cell growth have not been carefully elucidated.

We report here that SLIT2 prevents macrophage spreading not by inhibiting Rac activation, but rather, by activating RhoA. We further demonstrate that SLIT2 can inhibit macropinocytic uptake of bacterial PAMPs and subsequent upregulation of inflammatory chemokines in vivo, thereby modulating macrophage immune responses. We show that SLIT2, in a ROBO1-dependent manner, negatively impacts the survival of KRAS-transformed cancer cells by inhibiting macropinocytosis, thus limiting protein uptake in a Glut-deprived state similar to the tumor microenvironment found in vivo. Our work identifies an endogenous inhibitor of macropinocytosis with potent effects both in vitro and in vivo.

## Results

**NSlit2 induces actin remodeling in macrophages in a ROBO1-dependent manner.** We first investigated which Slit-binding Robo receptors are expressed in macrophages and found that *Robo1*, but not *Robo2*, messenger RNA (mRNA) is expressed in the RAW264.7 macrophage cell line and in primary murine bone marrow-derived macrophages (BMDM) (Fig. 1a). Using immunoblotting, ROBO1 protein was detected in RAW264.7 cells as well as in primary macrophages derived from human peripheral blood mononuclear (MDM) cells (Supplementary Fig. 1a). To investigate the effects of SLIT2 on macrophage cytoskeleton, we incubated cells with control vehicle, or with recombinant bioactive NSlit2 (the amino-terminal fragment of SLIT2), or with other recombinant SLIT2 proteins, CSlit2 and Slit2ΔD2, which do not bind to the Robo receptors (Fig. 1b)[22,30,32]. The endotoxin levels in all recombinant Slit protein preparations were below 0.05 EU/ml (Supplementary Fig. 1b; Supplementary Table 1). Treatment with NSlit2 significantly reduced macrophage spreading, as seen by reduction in their total three-dimensional surface area (Fig. 1c, d and Supplementary Fig. 1c). To quantify the circularity of cells, we calculated the 'shape factor' using Volocity 6.3 software[40]. Shape factor values range from 0 to 1, the latter defining a perfectly circular shape (see "Methods" for mathematical calculation). Only NSlit2 treatment significantly increased rounding in human and murine macrophages (Fig. 1c, e; Supplementary Fig. 1d). To determine whether these effects of NSlit2 were dependent on ROBO1, we used a specific siRNA to knockdown its expression in RAW264.7 cells (Fig. 1f; Supplementary Fig. 1e). NSlit2-induced inhibition of cell spreading was abolished by ROBO1 knockdown (Fig. 1g). Furthermore, in ROBO1-deficient macrophages, NSlit2 failed to induce the rounding (Fig. 1h). To validate this finding using an independent experimental approach, we pre-incubated NSlit2 with the soluble N-terminal fragment of the human ROBO1 receptor (Robo1N) containing the NSlit2-binding Ig1 domain[41]. Robo1N has previously been shown to act as an NSlit2 antagonist[22,42]. Treatment with Robo1N alone did not have any effect on cell spreading or rounding (Supplementary Fig. 1f, g). However, Robo1N-treated NSlit2 failed to reduce cell spreading (Supplementary Fig. 1f) as well as to increase cell rounding (Supplementary Fig. 1g). These

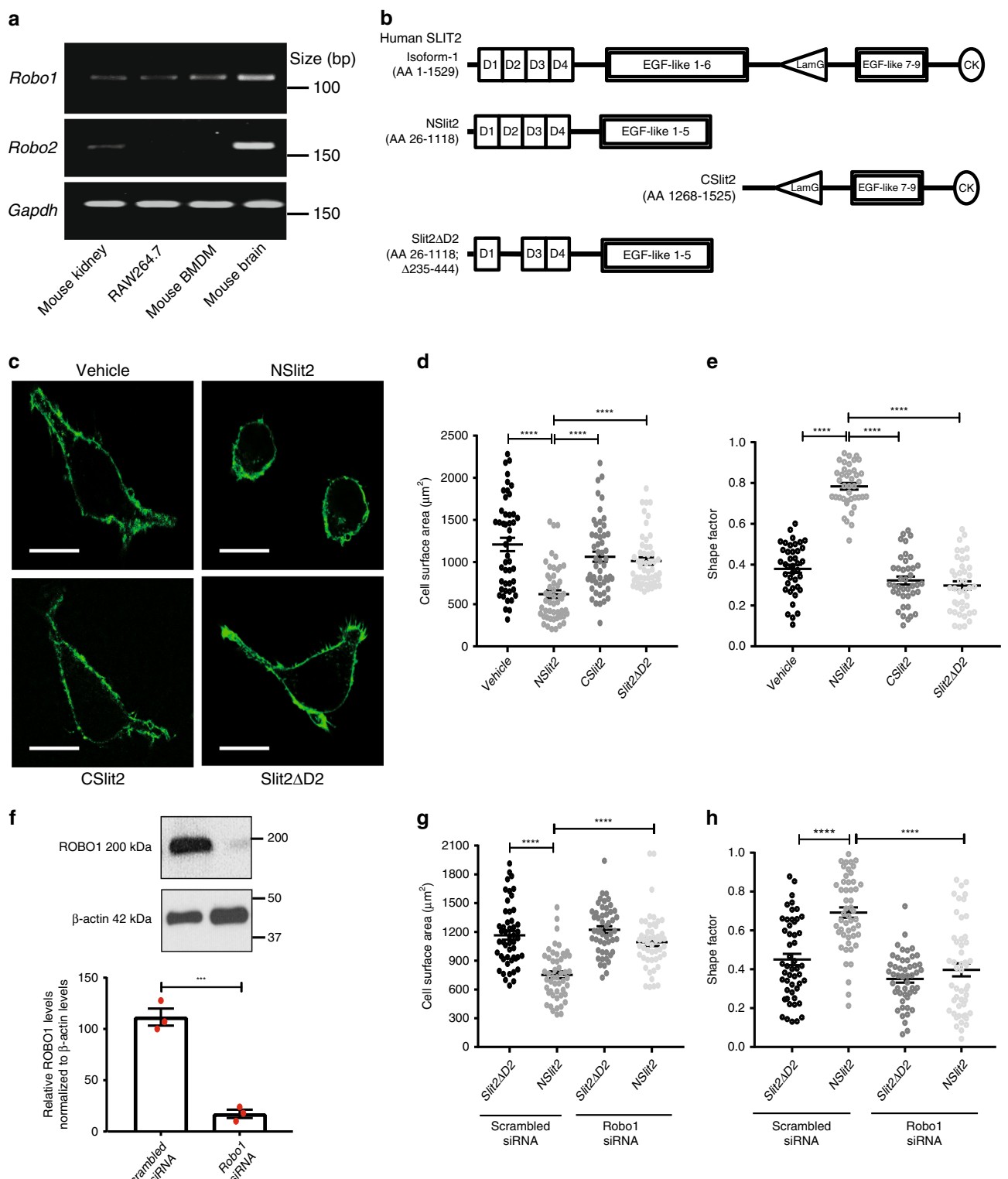

findings suggest that actions of NSlit2 on the macrophage cytoskeleton are mediated by its binding to the ROBO1 receptor.

SLIT2 has previously been shown to inhibit human and murine monocyte chemotaxis by inactivating Rac1[32]. In addition, it has been reported that NSlit2 can inactivate Rac1 and Cdc42 GTPases in RAW264.7 cells[43,44]. To determine whether inactivation of Rac1 is responsible for NSlit2-induced inhibition of macrophage spreading, we transiently expressed a constitutively active form of RAC1(Q61L)[45] in RAW264.7 cells. In line with previous reports[46,47], the Q61L mutant increased macrophage spreading (Supplementary Fig. 2a). The Q61L mutant, however, failed to prevent the decrease in cell spreading following exposure to NSlit2 (Supplementary Fig. 2a). To further validate this finding, we studied the effects of NSlit2 on primary BMDM from Rac1/2 double knockout (Rac1/2 DKO) mice. Rac1/2 DKO macrophages were smaller than the wild-type (WT) counterparts (Fig. 2a). However, exposure of Rac1/2 DKO macrophages to NSlit2 further reduced their spreading (Fig. 2a). The Arp2/3 protein

**Fig. 1 NSlit2 induces actin remodeling in macrophages in a ROBO1-dependent manner. a** Total RNA was isolated from murine brain and kidney, as well as from RAW264.7 cells and primary murine bone marrow-derived macrophages (BMDM). *Robo1* and *Robo2* mRNA expression were investigated by RT-PCR. *Gapdh* was used as a loading control. **b** A schematic representation of full-length human SLIT2 isoform 1 (accession number- NP_004778.1) protein, and recombinant NSlit2, CSlit2, and Slit2ΔD2 proteins used in this study. **c** RAW264.7 cells were incubated with vehicle (phosphate-buffered saline), NSlit2, CSlit2, or Slit2ΔD2 for 15 min at 37 °C and allowed to spread on poly-D-lysine-coated coverslips for 1 h. Cells were fixed, permeabilized, and incubated with AF-488-conjugated phalloidin (green) to label polymerized actin. Scale bar, 25 μm. **d, e, g, h** All data are presented as mean ± standard error of mean (SEM). Comparisons between the groups were made by one-way analysis of variance (ANOVA), followed by post hoc Tukey's multiple comparison test. $n = 50$ cells per treatment group per experiment over three independent experiments. **d** Experiments were performed as in (**c**) and cell surface area was measured using Volocity 6.3 software. ****$p < 0.0001$, NSlit2 vs vehicle, CSlit2, or Slit2ΔD2. **e** Shape factor for cells in (**c**) using Volocity 6.3 software. ****$p < 0.0001$, NSlit2 vs vehicle, CSlit2, or Slit2ΔD2. **f** ROBO1 levels were knocked down in RAW264.7 cells using specific siRNA. Protein levels (band intensity) were quantified using ImageJ software, version 1.51v and normalized to corresponding β-actin levels. $n = 3$ independent experiments (Supplementary Fig. 1e). Comparison between the two groups was made by unpaired, two-tailed t-test. ***$p = 0.0005$, scrambled vs Robo1 siRNA.
**g** ROBO1 protein levels were knocked down in RAW264.7 cells. After 72 h, experiments were performed as in (**c**). Cell surface area was measured as in (**d**). ****$p < 0.0001$ for the indicated comparisons and $p = 0.9887$, NSlit2 vs Slit2ΔD2 in ROBO1 knockdown conditions. (**h**) Shape factor for cells in (**g**) was determined as in (**e**). ****$p < 0.0001$ for the indicated comparisons and $p = 0.4268$, NSlit2 vs Slit2ΔD2 in ROBO1 knockdown conditions. Source data for (**a**, **d**, **e**, **f–h**) are provided as a Source Data file.

complex is an important downstream effector of both Rac1 and Cdc42 GTPases mediating actin polymerization[48]. Exposure to CK-666, a potent Arp2/3 inhibitor[49], could not reverse the NSlit2-induced cell spreading phenotype (Supplementary Fig. 2b). The results indicate that NSlit2-induced inhibition of macrophage spreading is independent of its actions on Rac1/2 and Cdc42 activities.

**NSlit2 activates RhoA and results in cytoskeletal rearrangement in macrophages.** We asked whether NSlit2 inhibited cell spreading through its effects on RhoA, which is the most abundantly expressed Rho-family GTPase member in monocytes/macrophages[50]. We used absorbance-based G-LISA assays to measure total and active RhoA levels in RAW264.7 cells. NSlit2 treatment significantly increased the active RhoA levels (Fig. 2b). We next tested the effects of TAT-C3 exoenzyme, a cell-permeable inhibitor of RhoA, RhoB, and RhoC, but not Rac1 nor Cdc42[51], on macrophage spreading. Cells incubated with TAT-C3 were significantly larger than the control counterparts (Fig. 2c, d). In macrophages treated with TAT-C3, NSlit2 no longer inhibited cell spreading (Fig. 2c, d). To verify this, we reduced the expression of RhoA using a siRNA (Supplementary Fig. 2c). Exposure to NSlit2 failed to attenuate spreading in RhoA-deficient cells (Supplementary Fig. 2d). Overall, these data indicate that NSlit2 inhibits the spreading of macrophages primarily through the activation of RhoA.

Several downstream effectors of RhoA, such as Rho-associated coiled-coil containing protein kinases (ROCK1/2) and diaphanous-related formins, are known to regulate actin polymerization via distinct mechanisms[52]. To determine which of these two pathways mediates NSlit2-induced cytoskeletal changes, we selectively blocked their actions. In the presence of Y-27632, a ROCK1/2 inhibitor[53], NSlit2 still inhibited macrophage spreading (Supplementary Fig. 2e). In contrast, SMIFH2, an inhibitor of formin homology 2 domains[54], completely abrogated the NSlit2-induced reduction in cell surface area (Fig. 2e; Supplementary Fig. 2f) and increase in the cell rounding (Fig. 2f; Supplementary Fig. 2f). These findings suggest that NSlit2 inhibits macrophage spreading primarily by augmenting RhoA-mediated formin activation.

**NSlit2-induced RhoA activation in macrophages is mediated by inactivation of MYO9B.** Backer S. et al. recently reported that NSlit2 activates TRIO, a dual Rho/Rac guanine nucleotide exchange factor (GEF), in murine embryonic fibroblasts and neuronal cells[55]. We found that *Trio* mRNA is expressed at low

levels in RAW264.7 cells but not in primary murine BMDM (Supplementary Fig. 3a). Liu et al. have shown that NSlit2 mediates RhoA activation, via inactivation of the Fyn kinase, a member of the Src-family tyrosine kinases, in rat oligodendrocyte precursor cells, but not in mature oligodendrocytes[56]. SLIT2-ROBO1-signaling has also been shown to mediate pan-Src activation in cancer cells[57,58]. However, NSlit2 failed to induce a significant inactivation of Fyn (Supplementary Fig. 3b) or activation of pan-Src kinases (Supplementary Fig. 3c) in RAW264.7 macrophages.

We next investigated the effects of NSlit2 on myosin IXb (MYO9B), an atypical motor protein, with multiple F-actin binding sites within its head (N-terminal) region and a RhoGAP domain in its tail (C-terminal) region[59]. Using immunoblotting, we confirmed higher endogenous MYO9B levels in RAW264.7 macrophages as compared to human embryonic kidney (HEK293T) cells (Supplementary Fig. 3d). The highly conserved RhoGAP domain of MYO9B alone can inactivate RhoA in cells[39], and we, therefore, transiently expressed the human MYO9B RhoGAP domain (MYO9B-GAP)-containing plasmid in RAW264.7 and HEK293T cells (Supplementary Fig. 3e). In the presence of bio-inactive Slit2ΔD2, the expression of MYO9B-GAP induced excessive spreading of macrophages (Fig. 3a, b), which was not reversed by the NSlit2 treatment (Fig. 3a, b). NSlit2-induced cell rounding was also prevented by MYO9B-GAP expression (Fig. 3a, c). In HEK293T cells stably over-expressing the ROBO1 receptor (HEK293T-ROBO1) but low endogenous MYO9B, NSlit2 did not affect cell spreading (Supplementary Fig. 3f) or rounding (Supplementary Fig. 3g) significantly. Upon exogenous expression of MYO9B-GAP in these cells, exposure to NSlit2 resulted in reduced spreading (Supplementary Fig. 3f) and increased rounding (Supplementary Fig. 3g). To verify our findings, we incubated cells with blebbistatin, an inhibitor of the myosin II family of proteins, which reverses the MYO9B dominant negative phenotype[60]. Blebbistatin treatment did not completely reverse NSlit2-induced inhibition of cell spreading (Fig. 3d, e) but prevented NSlit2-induced cell rounding (Fig. 3d, f). Together, these findings suggest that the SLIT2-ROBO1-MYO9B-RhoA pathway results in decreased cell spreading and increased cell rounding in macrophages.

**NSlit2 inhibits constitutive and induced macropinocytosis in vitro and in vivo.** We next investigated the effects of NSlit2 on macrophage macropinocytosis, a process dependent on the actin cytoskeleton[61]. Macropinosomes are defined as endocytic vacuoles larger than 0.2 μm in diameter[8]. As a first step, we verified the

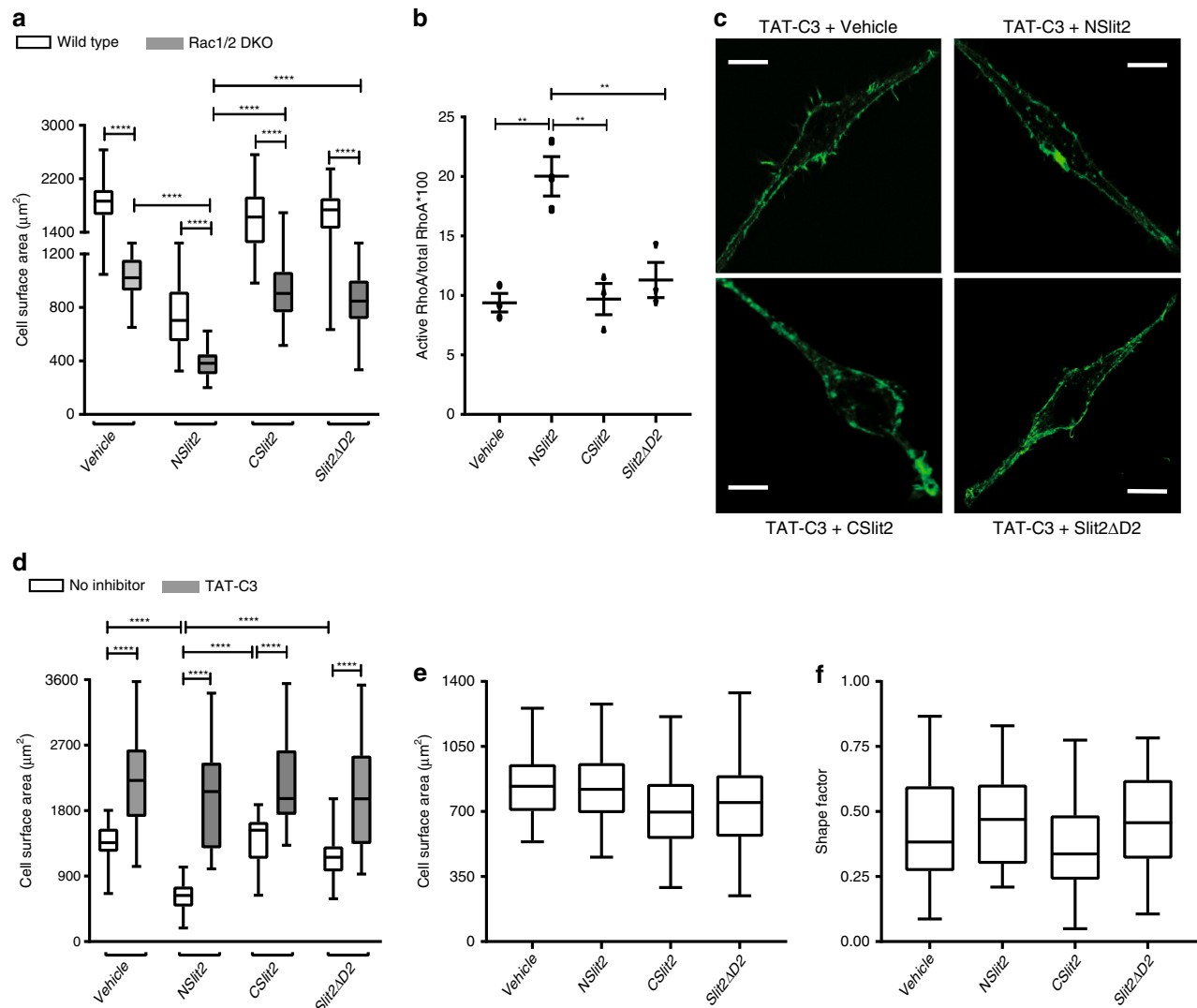

**Fig. 2 NSlit2 activates RhoA and results in cytoskeletal rearrangement in macrophages.** (**a**, **d**, **e**, **f**) Data are presented as boxplots where middle line is the median; lower and upper hinges correspond to the first and third quartiles; whiskers represent minimum and maximum values. $n = 50$ cells per treatment group per experiment over three independent experiments. Comparisons between the groups were made by one-way ANOVA in (**b**, **e**, **f**) and two-way ANOVA in (**a**, **d**), followed by post hoc Tukey's multiple comparison test. **a** Primary BMDM from wild-type (WT) or Rac1/2 double knockout (DKO) mice were cultured for 10 days, and cell spreading assays were performed as described in Fig. 1c. Cell surface area was measured as in Fig. 1d. ****$p < 0.0001$, WT vs DKO BMDM and $p < 0.0001$, NSlit2 vs vehicle, CSlit2, or Slit2ΔD2. **b** RAW264.7 macrophages were incubated with vehicle, NSlit2, CSlit2 or Slit2ΔD2 for 15 min at 37 °C and active and total RhoA levels were measured using a G-LISA assay. **$p = 0.0063$, 0.0093, and 0.0053 for NSlit2 vs vehicle, CSlit2, Slit2ΔD2, respectively. **c** RAW264.7 cells were incubated with the RhoA/B/C inhibitor, TAT-C3, for 4 h followed by vehicle, NSlit2, CSlit2, or Slit2ΔD2 treatment for 15 min and cell spreading was performed as described in Fig.1c. Scale bar, 20 μm. **d** Surface area for cells in (**c**) was measured as in Fig. 1d. ****$p < 0.0001$, vehicle vs TAT-C3 and $p = 0.8877$, 0.9470, and 0.9952 for NSlit2 vs vehicle, CSlit2, Slit2ΔD2 in TAT-C3-treated conditions, respectively. **e**, **f** Experiments were performed as in Fig. 1c but cells were first incubated with a formin inhibitor, SMIFH2, for 30 min. **e** Cell surface area was measured as in Fig. 1d. $p = 0.9220$, 0.8356, and 0.7215 for NSlit2 vs vehicle, CSlit2, Slit2ΔD2, respectively. **f** Shape factor was measured as in Fig. 1e. $p = 0.7557$, 0.6594, and 0.9950 for NSlit2 vs vehicle, CSlit2, Slit2ΔD2 respectively. Source data for (**a**, **b**, **d**, **e**, **f**) are provided as a Source Data file.

effects of modulating RhoA signaling on macropinocytosis. Inhibition of RhoA/B/C, using TAT-C3, induced a twofold increase in constitutive macropinocytosis by RAW264.7 macrophages (Fig. 4a, b). Conversely, activation of RhoA/B/C using Rho Activator II significantly decreased average number of macropinosomes per cell (Fig. 4a, b). There were also changes in the average macropinosome size with TAT-C3 and Rho Activator II treatments (Fig. 4c). Next, we examined the effect of NSlit2 on constitutive macropinocytosis. Exposure to NSlit2 significantly reduced the average number of macropinosomes per cell (Fig. 4d, e) but did not produce a statistically significant change in the macropinosome size (Fig. 4f). In RhoA-deficient macrophages, constitutive macropinocytosis was

significantly enhanced (Supplementary Fig. 4a, b). RhoA knockdown partially, but significantly, restored macropinocytosis after exposure to NSlit2 (Supplementary Fig. 4a, b). To explain this result, we investigated the specific effect of NSlit2-induced Rho activation on its inhibition of Rac activity[43]. As expected, pharmacological inhibition of Rho activity with TAT-C3 resulted in an increase in total active Rac levels in macrophages, as measured by Rac1/2/3 G-LISA (Supplementary Fig. 4c). NSlit2 treatment, following TAT-C3, further reduced the Rac activity (Supplementary Fig. 4c). These findings suggest that NSlit2 inhibits macropinocytosis in vitro by activation of RhoA, as well as through inactivation of Rac1, possibly via SRGAP2, a Rac1-specific slit-robo

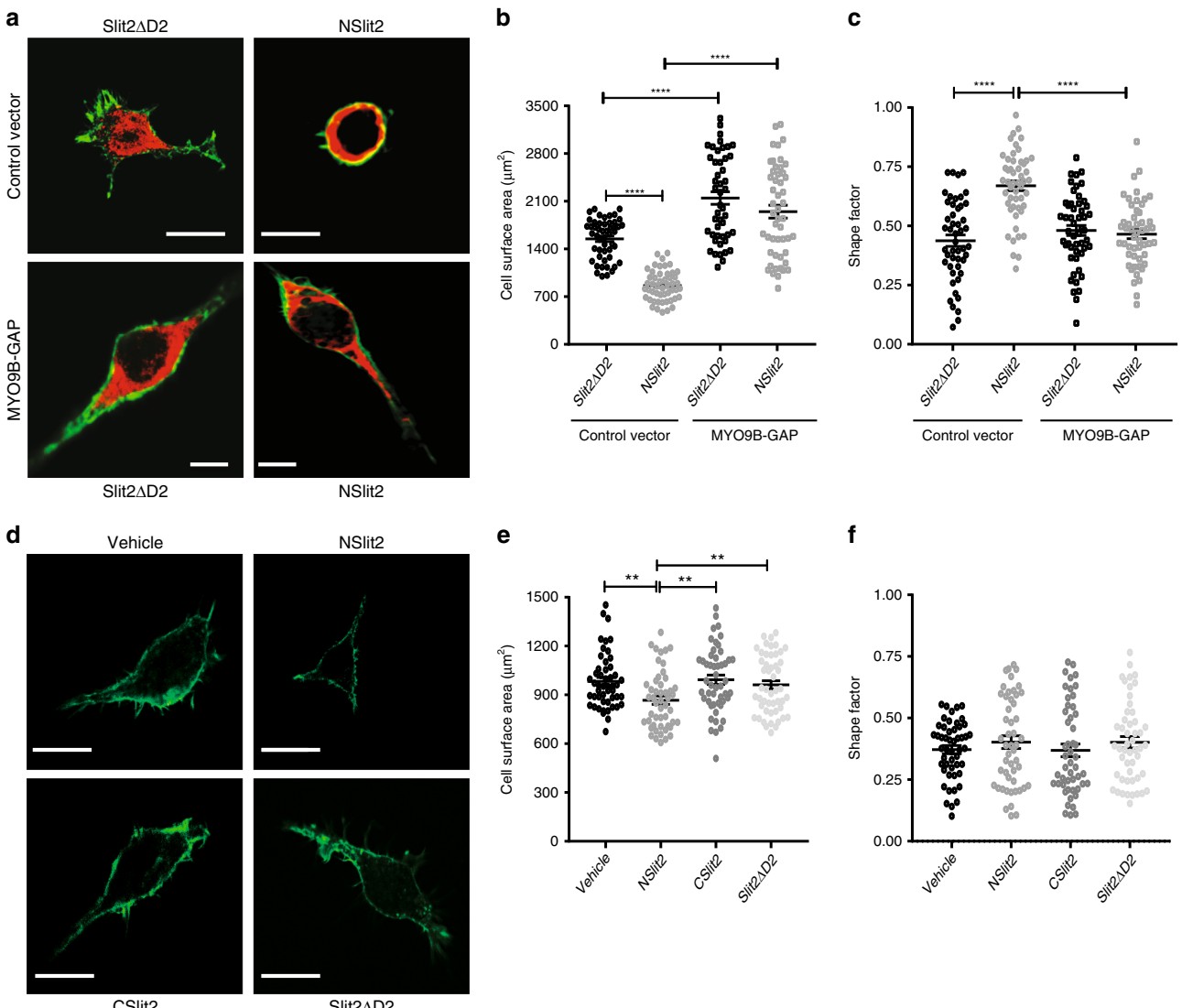

**Fig. 3 NSlit2-induced RhoA activation in macrophages is mediated by inactivation of MYO9B. a** A cDNA plasmid encoding c-myc-tagged MYO9B-GAP (Supplementary Fig. 3e) was expressed in RAW264.7 cells. After 48 h, cells were incubated with NSlit2 or Slit2ΔD2 for 15 min at 37 °C and allowed to spread on poly-ᴅ-lysine-coated coverslips for 1 h. Cells were fixed, permeabilized, and incubated with AF-488-conjugated phalloidin (green) and an antibody directed to c-Myc (red). Scale bar, 25 μm. (**b**, **c**, **e**, **f**) Data are presented as mean ± SEM. Comparisons between the groups were made by ANOVA, followed by post hoc Tukey's multiple comparison test. $n = 50$ cells per treatment group per experiment over three independent experiments. **b** Experiments were performed as in (**a**). Cell surface area was measured as in Fig. 1d. ****$p < 0.0001$, for the indicated comparisons and $p = 0.3388$, MYO9B-GAP NSlit2 vs MYO9B-GAP Slit2ΔD2. **c** Shape factor for cells in (**a**) was measured as in Fig. 1e. ****$p < 0.0001$, for the indicated comparisons and $p = 0.6416$, MYO9B-GAP NSlit2 vs MYO9B-GAP Slit2ΔD2. **d** Experiments were performed as Fig. 1c but cells were first incubated with the myosin II inhibitor, blebbistatin, for 30 min. Scale bar, 25 μm. **e** Cell surface area was measured for (**d**) as in Fig. 1d. **$p = 0.0032$, 0.0043, and 0.0022 for NSlit2 vs vehicle, CSlit2, and Slit2ΔD2, respectively. **f** Shape factor for cells in (**d**) was measured as in Fig. 1e. $p = 0.7848$, 0.5049, and 0.6879 for NSlit2 vs vehicle, CSlit2, and Slit2ΔD2, respectively. Source data for (**b**, **c**, **e**, **f**) are provided as a Source Data file.

GTPase activating protein (srGAP)[62], expressed in macrophages (Supplementary Fig. 4d).

We asked whether NSlit2 could inhibit inducible macropinocytosis in primary BMDM incubated with CSF1[3,5]. Exposure to NSlit2 also blocked CSF1-induced macropinocytosis in BMDM (Supplementary Fig. 4e-f). We next examined whether NSlit2 could prevent macrophage macropinocytosis in vivo. We previously reported that administration of recombinant NSlit2 inhibits monocyte/macrophage recruitment to the peritoneal cavity in a murine model of sterile peritonitis[32]. To test whether NSlit2 could suppress macropinocytosis in this model, 48 h after induction of peritonitis to allow for the initial monocyte/macrophage migration[63], we administered NSlit2 followed by

intraperitoneal injection of fluorescently labeled dextran (Fig. 4g). Macrophages were identified by immunofluorescent labeling of CD68 (Supplementary Fig. 4g). Mice treated with NSlit2 displayed significantly less macropinocytosis by the peritoneal macrophages compared to mice treated with vehicle, CSlit2, or Slit2ΔD2 (Fig. 4h, i). Together, these findings indicate that NSlit2 inhibits macropinocytosis by macrophages in vitro and in vivo.

**NSlit2 inhibits MDP uptake and NOD2-mediated NF-κB activation in macrophages.** Constitutive macropinocytosis in macrophages and iDCs has been implicated in the uptake of soluble antigens from the extracellular fluid[4,8]. MDP, a peptidoglycan motif

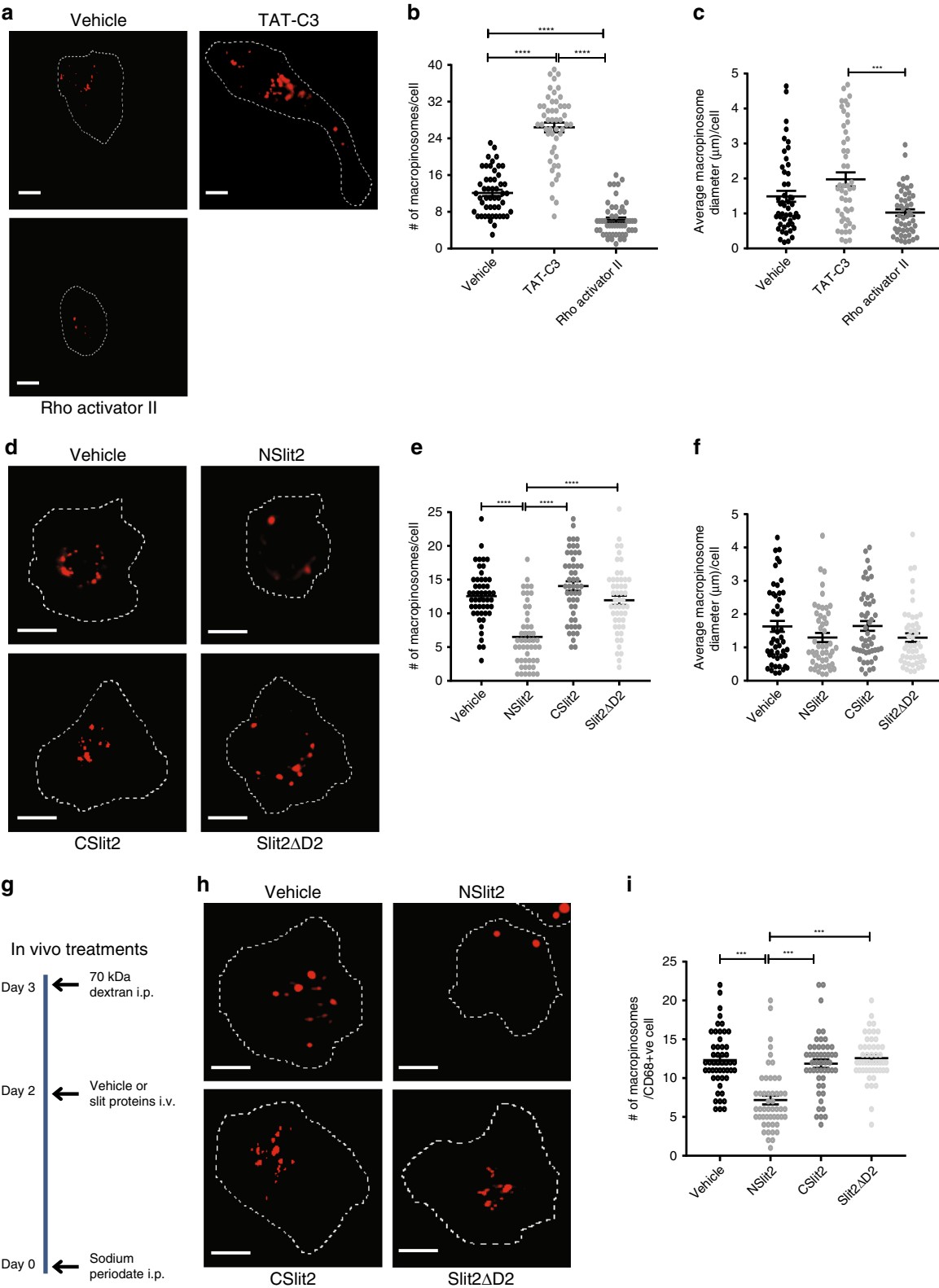

found in bacteria, is taken up by primary human macrophages via macropinocytosis[5], and binds to the intracellular Nucleotide-binding oligomerization domain containing 2 (NOD2) receptor. MDP binding to NOD2 induces NF-κB-mediated secretion of pro-inflammatory cytokines, including the chemokine, C-X-C motif ligand 1 (CXCL1)[64]. We found that upon exposure to NSlit2, intake of FITC-conjugated MDP (MDP-FITC) by primary BMDM was

significantly reduced (Fig. 5a, b). We next assessed the effects of NSlit2 on MDP-induced NF-κB activation. In the absence of MDP, NSlit2 did not affect NF-κB activity, as assessed by measuring the activation of the p65 (phospho-Ser536) subunit of NF-κB (Fig. 5c, d). However, NSlit2, but not Slit2ΔD2, blocked MDP-induced NF-κB activation (Fig. 5c, d). These findings indicate that NSlit2 inhibits macropinocytosis, thereby attenuating NOD2-mediated NF-κB

**Fig. 4 NSlit2 inhibits constitutive and induced macropinocytosis in vitro and in vivo. a** RAW264.7 cells were treated with RhoA/B/C inhibitor, TAT-C3, or Rho Activator II for 4 h, then incubated with Tetramethylrhodamine (TMR)-labeled 70 kDa dextran (red) for 15 min at 37 °C. Cells were washed and fixed. Bright-field images were captured and cell boundaries are demarcated with dashed lines. Scale bar, 20 μm. (**b, c, e, f**) macropinosomes (number and size) were measured using ImageJ software, version 1.51v. All data are presented as mean ± SEM. Comparisons between groups were made by Kruskal–Wallis ANOVA, followed by Dunn's multiple comparisons test. $n = 50$ cells per treatment group per experiment over three independent experiments. **b** Macropinosomes were counted for cells in (**a**). ****$p < 0.0001$, vehicle vs TAT-C3 and Rho Activator II vs vehicle or TAT-C3. **c** Macropinosome diameters were measured for cells in (**a**). ***$p = 0.0002$, TAT-C3 vs Rho Activator II. **d** RAW264.7 cells were incubated with vehicle, NSlit2, CSlit2, or Slit2ΔD2 for 15 min at 37 °C and macropinocytosis assays performed as described in (**a**). Scale bar, 10 μm. **e** Macropinosomes were counted for cells in (**d**). ****$p < 0.0001$, NSlit2 vs vehicle, CSlit2, or Slit2ΔD2. **f** Macropinosome diameters were measured for cells in (**d**). $p = 0.1913$, 0.2509, and 0.4802 for NSlit2 vs vehicle, CSlit2, Slit2ΔD2, respectively. **g** Schematic of in vivo treatments to investigate macropinocytosis. Mice were injected intraperitoneally (i.p.) with sodium periodate to induce peritonitis. After 48 h, saline vehicle, NSlit2, CSlit2, or Slit2ΔD2 was administered intravenously. After 18 h, TMR-labeled 70 kDa dextran (red) was administered i.p. After 30 min, mice were euthanized, and peritoneal macrophages were isolated, attached to poly-D-lysine-coated coverslips, fixed and incubated with FITC-conjugated anti-CD68 Ab. **h** In vivo macropinocytosis was performed as described in (**g**). $n = 6$ mice per treatment group. Scale bar, 10 μm. **i** Macropinosomes were counted for cells in (**h**). Thirty cells per animal per treatment group were analyzed. ***$p = 0.0006$, 0.0003, 0.0008 for NSlit2 vs vehicle, CSlit2, Slit2ΔD2 respectively. Source data for (**b, c, e, f, i**) are provided as a Source Data file.

activation in macrophages by preventing the cellular entry of NOD2 ligands. We next examined the effects of NSlit2 on NOD2-induced secretion of CXCL1 in a mouse model of sterile peritonitis[32,63]. After the exposure of peritoneal macrophages to MDP, CXCL1 can be detected in serum as early as 2 h[64]. We found that treatment with either NSlit2 or the pharmacological macropinocytosis inhibitor, 5-(N-Ethyl-N-isopropyl)amiloride (EIPA)[65], reduced the MDP-induced rise in serum CXCL1 levels (Fig. 5e). In line with our previously published results[32], NSlit2 also diminished the MDP-induced influx of immune cells into the peritoneal cavity (Supplementary Fig. 5a). Remarkably, even after normalizing for the number of cells in the peritoneal cavity, NSlit2 attenuated the levels of CXCL1 detected in the serum (Fig. 5f), suggesting that this effect of NSlit2 is independent of its action on cell migration. Both NSlit2 and EIPA also reduced the levels of CXCL1 detected locally in the peritoneal exudate (Supplementary Fig. 5b). These results demonstrate that exogenously administered NSlit2 potently inhibits MDP signaling in vivo by inhibiting macropinocytosis.

We next sought to understand how endogenous SLIT proteins regulate macropinocytosis. As a first step, we used a recently validated ELISA kit[66,67] to measure the SLIT2 protein levels in the serum and peritoneal membrane samples of adult C57BL6/J mice. The levels of endogenous SLIT2 in the peritoneum were ~10 times higher than those detected in serum (Fig. 5g). Because two recent studies have reported that SLIT3 protein is enriched in some extra-neuronal peripheral tissues such as bone marrow, we used a validated ELISA kit[68] to measure SLIT3, and found that SLIT3 levels in murine serum and peritoneal membrane samples are similar (Supplementary Fig. 5c)[24,68]. We found that SLIT2 levels in peritoneal membrane samples are significantly higher than those of SLIT3 (Supplementary Fig. 5d).

We next tested the effect of blocking endogenous SLIT2 on MDP signaling by administering a single intraperitoneal dose of soluble Robo1N. Robo1N itself did not upregulate CXCL1 (Fig. 5h) but significantly augmented MDP-induced CXCL1 in serum (Fig. 5h) as well as in the peritoneal fluid (Supplementary Fig. 5e). The effects of Robo1N and MDP were reversed by the macropinocytosis inhibitor, EIPA (Fig. 5h and Supplementary Fig. 5e). The results demonstrate that endogenous SLIT2 is a physiological inhibitor of macropinocytosis.

**NSlit2 attenuates protein uptake by macropinocytosis in cancer cells and restricts their growth in a ROBO1-dependent manner.** Macropinocytosis is an important nutritional adaptation to maintain the supply of amino acids, especially Glut, in RAS-transformed tumor cells in vitro and in vivo[11,17]. In

PDAC, low levels of *SLIT2* mRNA are associated with worse prognosis in patients[38]. We, therefore, asked if SLIT2 could inhibit macropinocytosis, and therefore the growth of RAS-transformed cancer cells. It has previously been reported that PDAC-dervied PANC-1 cells express the ROBO1 receptor[38] but intestinal adenocarcinoma-derived DLD-1 cells do not;[69,70] even though both cell lines have oncogenic KRAS mutations and perform robust macropinocytosis[11,13]. We first tested whether these cell lines depend on macropinocytosis for protein intake and for their survival, when Glut availibility in the extracellular media is restricted to levels typically found in the tumor microenvironment[11]. Cell viability was reduced by more than fourfold when grown in Glut-deprived culture medium (Fig. 6a). This effect could be reversed by addition of protein in the form of 2% albumin to the medium (Fig. 6a), but was blocked by treatment with the macroinocytosis inhibitor, EIPA (Fig. 6a). NSlit2 treatment significantly reduced albumin uptake in ROBO1-expressing PANC-1 cells (Fig. 6b) but not in ROBO1-deficient DLD-1 cells (Fig. 6c). Finally, the addition of NSlit2 to the medium significantly decreased the viability of PANC-1 cells (Fig. 6d) but no such effect was observed for DLD-1 cells (Fig. 6e). These findings suggest that NSlit2 is a potent inhbitor of macropinocytosis in KRAS-transformed tumor cells, in a ROBO1-dependent manner.

## Discussion

SLIT2 protein is widely expressed in adult, non-neuronal tissues[24,25], but its actions on tissue-resident immune cells, such as macrophages, remain largely unexplored. We report here that primary murine and human macrophages express a single NSlit2 receptor, ROBO1. These results are in keeping with recent work showing that murine BMDM express ROBO1[24]. We found that exposure to bioactive NSlit2, but not to the bio-inactive Slit2ΔD2 peptide, inhibited macrophage spreading and induced cell rounding, in a ROBO1-dependent manner. ROBO1 and SLIT2 global knockout mice fail to survive beyond 3 weeks[71,72], and hypomorphic deletion of ROBO1 in mice results in osteopenia[24]. To inhibit SLIT2/ROBO1 signaling, we, therefore, used soluble Robo1N, which contains the Slit-binding Ig1 domain of the human ROBO1 receptor, and which has been previously shown to act as an NSlit2 antagonist[42]. Our findings add to a body of work demonstrating that SLIT2/ROBO1 signaling significantly impacts the rearrangement of the actin cytoskeleton in immune cells[30,73].

We observed that the effects of NSlit2 to inhibit macrophage spreading and to induce cell rounding did not occur through

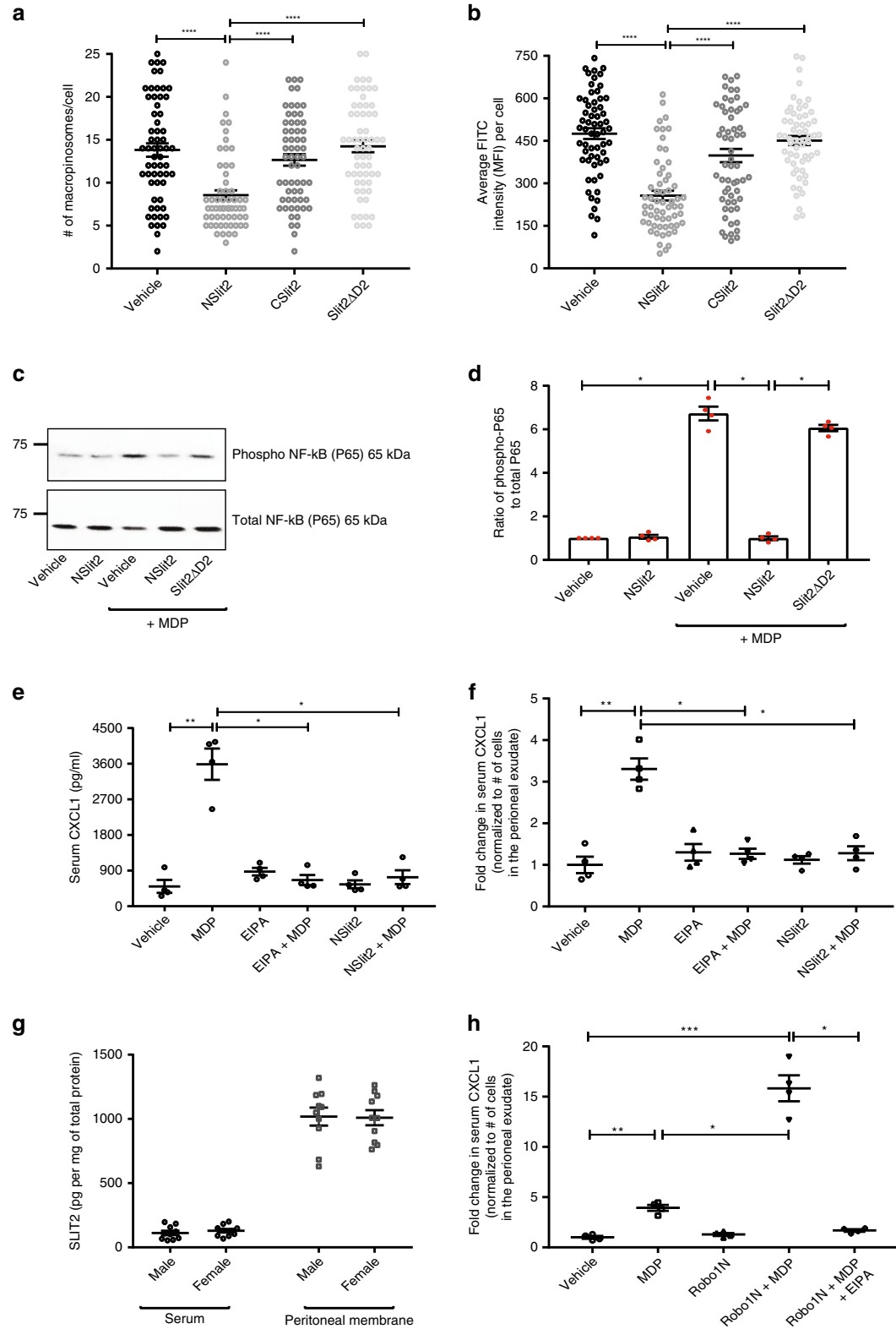

inactivation of Rac1/2, but instead, through activation of RhoA. These results are in contrast to the previous studies demonstrating that NSlit2 binding to ROBO1 influences actin dynamics by recruiting and activating GAPs, including srGAPs, that in turn inhibit activation of Rac1/2 and Cdc42[32,73–75]. Our group previously reported that SLIT2, acting via ROBO1, inhibits monocyte chemotaxis by preventing chemokine-induced activation of Rac1 and Cdc42 GTPases. In line with those results, in the present study, we found that NSlit2 treatment attenuated MDP-induced immune cell migration into the peritoneal cavity (Supplementary Fig. 5a). We previously demonstrated that SLIT2 inhibits post-adhesion stabilization of monocytes tethered to the activated endothelial cells[32]. Our group has also reported that NSlit2 impairs neither phagocytosis nor reactive oxygen species production in neutrophils-cellular processes dependent on activation of Rac and Cdc42 and importantly, that NSlit2-treated mice clear

**Fig. 5 NSlit2 inhibits MDP uptake and NOD2-mediated NF-κB activation in macrophages.** (**a**, **b**, **d**–**h**) All data are presented as mean ± SEM. Comparisons between groups were made by Kruskal–Wallis ANOVA, followed by Dunn's multiple comparisons test. **a**, **b** Macropinosome analysis was performed using ImageJ software, version 1.51v. $n = 50$ cells per treatment group per experiment over three independent experiments. **b** BMDM were incubated with vehicle, NSlit2, CSlit2, or Slit2ΔD2 for 15 min, then with FITC-conjugated muramyl dipeptide (MDP) for 30 min. ****$p < 0.0001$, NSlit2 vs vehicle, CSlit2, or Slit2ΔD2. **b** Experiments were performed as in (**a**). MFI of the FITC channel was measured. ****$p < 0.0001$, NSlit2 vs vehicle, CSlit2, or Slit2ΔD2. **c** Experiments were performed as described in (**a**). Total protein lysates were collected 5 min after MDP treatment. **d** Quantification of the phospho-p65/total-p65 ratio from 4 independent experiments as described in (**c**) using ImageJ software, version 1.51v. *$p = 0.0278$, MDP vs vehicle; *$p = 0.0192$, NSlit2 + MDP vs MDP alone; *$p = 0.0272$ NSlit2 + MDP vs Slit2ΔD2 + MDP; and $p > 0.9999$, Slit2ΔD2 + MDP vs MDP alone. **e** Mice were injected i.p. with sodium periodate to induce sterile peritonitis. After 72 h, saline vehicle, NSlit2, or EIPA was administered i.p. One hour later, MDP was administered i.p. After 2 h, blood and peritoneal exudates were collected. Serum CXCL1 levels were determined by ELISA. $n = 4$ animals per treatment group. **$p = 0.0048$, MDP vs vehicle; *$p = 0.0219$, NSlit2 + MDP vs MDP alone; and *$p = 0.0122$, EIPA + MDP vs MDP alone. **f** Experiments were performed as described in (**e**). Fold changes in serum CXCL1 were normalized for total number of cells in the peritoneal exudates. **$p = 0.0019$, MDP vs vehicle; *$p = 0.0455$, NSlit2 + MDP vs MDP alone; *$p = 0.0388$, EIPA + MDP vs MDP alone. **g** Serum samples were collected from mice as described in (**e**). Peritoneal membranes were collected in cold PBS. SLIT2 protein levels were measured using ELISA. $n = 10$ animals per group. **h** Sterile peritonitis was induced in mice as in (**e**). After 72 h, vehicle, Robo1N, or EIPA and Robo1N together was administered i.p. One hour later, MDP was administered i.p. After 2 h, blood and peritoneal exudates were collected. Fold changes in serum CXCL1 were normalized as described in (**f**). $n = 4$ animals per treatment group. ***$p = 0.0003$, Robo1N + MDP vs vehicle, **$p = 0.0086$, MDP vs vehicle; *$p = 0.0486$, Robo1N + MDP vs MDP alone; and *$p = 0.0365$, Robo1N + MDP vs. Robo1N + MDP + EIPA. Source data for (**a**–**h**) are provided as a Source Data file.

bacteria as effectively as their vehicle-treated counterparts[31]. NSlit2 does not affect Rac1 and Cdc42 activities in human and murine platelets[22]. Le et al. recently reported that NSlit2 interacts with ROBO1 to enhance contraction of mammary myoepithelial cells and their consequent pulling on the extracellular matrix by activating Rac1[76]. Similarly, in retinal vascular endothelial cells, SLIT2 signaling through ROBO1 and ROBO2 also activates Rac1 and promotes lamellipodial formation[23]. Overall, accumulating evidence suggests that differences in phenotypic effects and the signaling pathways transduced after exposure to NSlit2 are dependent on the cell type, the tissue milieu, and the biologic context.

SLIT2/ROBO1 signaling has been reported to activate RhoA in other cell types by either inactivation of Fyn kinase[56], or activation of TRIO[55]. SLIT2 can mediate RhoA inhibition via calcium signaling in neurons[77]. Following a brief exposure to NSlit2, we observed a significant increase in RhoA activity (Fig. 2b), and yet the same treatment failed to inactivate Fyn (Supplementary Fig. 3b). Our results are in keeping with the previous study which observed Fyn inactivation only after prolonged exposure to SLIT2[56]. In line with another study[78], we detected *Trio* mRNA in transformed RAW264.7 cells but not in primary BMDM (Supplementary Fig. 3a). This suggests that SLIT2-induced stimulation of TRIO might contribute to RhoA activation in RAW264.7 cells but cannot explain the phenotype we observed in primary macrophages.

We report here that in macrophages, NSlit2 binding to ROBO1 leads to cytoskeletal rearrangement by inactivating MYO9B, thereby promoting activation of RhoA, the most abundantly expressed Rho-family member in monocytes and macrophages[50]. Our results are in line with a study by Geutskens et al. reporting that SLIT3 activates RhoA in monocytes[79]. However, the mechanism of SLIT3-induced RhoA activation remains to be elucidated. Our findings are consistent with recent work demonstrating that exposure to NSlit2 causes inactivation of MYO9B in human lung carcinoma cells[39], and inactivation of the MYO9B ortholog in *Caenorhabditis elegans*[80]. The present study's findings are in contrast to our previous observation that, in rat fibroblasts, NSlit2 inhibited transforming growth factor-β-induced actin stress fiber formation, possibly by inhibiting RhoA[70]. In Schwann cells, SLIT2-induced RhoA (and ROCK) activity is needed for reversal of soma translocation but not for the collapse of the leading front[81]. Differential effects of NSlit2 to activate or inactivate RhoA/Rac1/Cdc42 in different cell types

may depend on the relative expression of the Rho-family GTPases as well as the Slit/Robo effector proteins, such as srGAPs and MYO9B. Notably, MYO9B is ubiquitously expressed in immune cells, with particularly high expression in monocytes and macrophages[82]. Accordingly, we found that HEK293T cells, which express low levels of endogenous MYO9B (Supplementary Fig. 3d), only become rounded in response to NSlit2 after transient expression of MYO9B RhoGAP domain (Supplementary Fig. 3g), which inactivates RhoA in transfected cells[39]. A recent study implicates MYO9B in the regulation of constitutive activity of RhoA to generate self-limiting cellular contraction patterns in the human osteosarcoma cell line, U2OS[83]. If this function of MYO9B is conserved in macrophages remains to be elucidated. To determine whether NSlit2-induced activation of RhoA results in downstream activation of Rho kinases, formins and/or myosin II, we used selective inhibitors of each of these pathways. Rock1/2 inhibition had no demonstrable effects, whereas formin inhibition reversed NSlit2's effects on both macrophage spreading and rounding. These data are in keeping with recent reports that formins can initiate actomyosin contraction[84,85]. Furthermore, MYO9B-deficient murine macrophages are also round in morphology and spread less after treatment with Rho kinase inhibitor, suggesting a role of formins in the regulation of macrophage spreading[82]. Finally, inhibition of myosin II activity in the presence of NSlit2 led to attenuation of macrophage spreading, but did not induce cell rounding. The exact role of myosin II in cell spreading is not clear. In murine fibroblasts, nonmuscle myosin IIa inhibits spreading by increasing retrograde F-actin flow[86]. However, recent work by Oakes et al. suggests that cell spreading is regulated by forces not generated by myosin motors[87]. Our results suggest that RhoA-mediated formin activation is necessary for NSlit2-induced actomyosin changes and that NSlit2-induced myosin II activation is needed for cell rounding but not for spreading.

Macrophages and iDCs sample their surroundings via macropinocytosis, which can be either constitutive or induced by growth factors. Both types require acute changes in the cortical cytoskeleton which are brought about by interactions between membrane phosphoinositides and active forms of small Rho-family GTPases, such as Rac1 and Cdc42[3,8]. We demonstrate here that activation of RhoA signaling suppresses constitutive macropinocytosis of macrophages. In line with this observation, exposure to NSlit2 potently inhibited macropinocytosis. We observed that RhoA-deficient macrophages exhibit limited

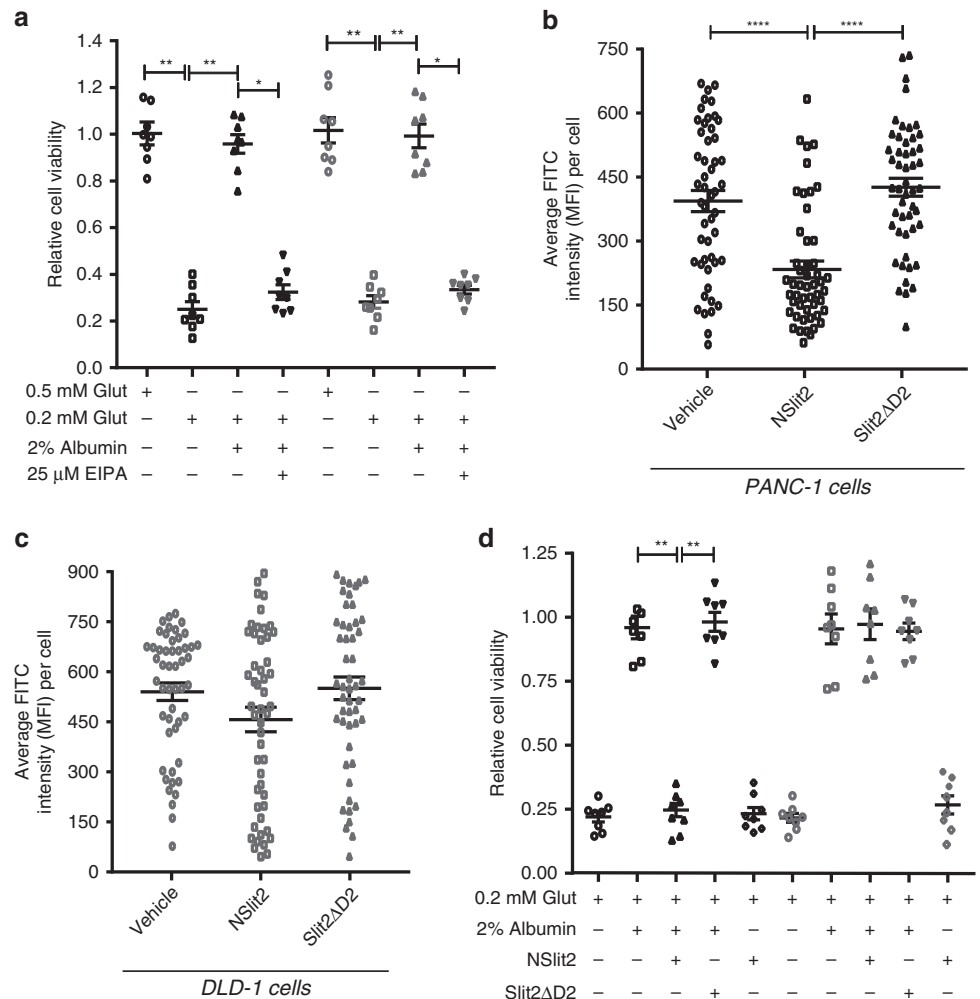

**Fig. 6 NSlit2 attenuates protein uptake by macropinocytosis in cancer cells and restricts their growth in a ROBO1-dependent manner. a–d** All data are presented as mean ± SEM. Comparisons between the groups were made by Kruskal–Wallis ANOVA, followed by Dunn's multiple comparisons test. PANC-1 and DLD-1 treatments are represented by black and gray symbols, respectively. **a, d** $n = 8$ independent replicates per treatment group per experiment over three independent experiments. **a** Oncogenic KRAS-expressing PANC-1 and DLD-1 cells were grown for 6 days in glutamine-free medium supplemented with high glutamine (Glut) (0.5 mM), low Glut (0.2 mM), 2% albumin. The macropinocytosis inhibitor, EIPA, was added, where indicated, for the last 2 days. Cell viability was measured on Day-6 using a MTT assay kit. Data are represented relative to the values obtained for 0.5 mM Glut supplementation. **$p = 0.0015$ and $0.0012$, 0.5 mM vs 0.2 mM Glut for PANC-1 and DLD-1 cells, respectively; **$p = 0.0013$ and $0.0036$, 0.2 mM Glut vs 0.2 mM Glut + 2% Alb for PANC-1 and DLD-1 cells, respectively; and *$p = 0.0276$ and $0.0360$, 0.2 mM Glut + 2% Albumin vs 0.2 mM Glut + 2% Albumin + 25 μM EIPA for PANC-1 and DLD-1 cells, respectively. **b, c** $n = 50$ cells per treatment group per experiment over three independent experiments. **b** PANC-1 cells were incubated with vehicle, NSlit2, or Slit2ΔD2 for 15 min at 37 °C and macropinocytosis assays performed as described in Fig. 4a using Albumin-FITC in place of the TMR-labeled dextran. Mean fluorescence intensity (MFI) of the FITC channel per cell was measured using ImageJ software, version 1.51 v. ****$p < 0.0001$, NSlit2 vs vehicle, or Slit2ΔD2. **c** Experiments were performed as in (**b**) using DLD-1 cells instead of PANC-1 cells. $p = 0.4469$ and $0.2049$, NSlit2 vs vehicle or Slit2ΔD2, respectively. (**d**) Cell viability assays were performed as described in (**a**) with addition of 0.2 mM glutamine, 2% albumin, NSlit2 and Slit2ΔD2 to the medium for 6 days, as indicated. **$p = 0.0061$ and >0.9999, 0.2 mM Glut + 2% Albumin vs 0.2 mM Glut + 2% albumin + NSlit2 for PANC-1 and DLD-1 cells, respectively. **$p = 0.0048$ and >0.9999, 0.2 mM Glut + 2% Albumin + NSlit2 vs 0.2 mM Glut + 2% Albumin + Slit2ΔD2 for PANC-1 and DLD-1 cells, respectively. Source data for (**a–d**) are provided as a Source Data file.

inhibition of macropinocytosis in the presence of NSlit2 (Supplementary Fig. 4b), which may reflect partial, rather than complete knockdown of RhoA. Another possible explanation is that in RhoA-deficient macrophages, NSlit2 could inhibit macropinocytosis by preventing activation of Rac1 (Supplementary Fig. 4c). Accordingly, we found SRGAP2, a Rac1-specific GAP that acts as an effector of Slit-Robo signaling[88,89], is highly expressed in murine macrophages. Chabaud et al. recently proposed that constitutive macropinocytosis and tissue infiltration by iDCs are mutually exclusive, and that both processes are regulated by cellular localization of nonmuscle myosin IIa[90]. In contrast, we found that SLIT2 negatively regulates migration[43] as well as macropinocytosis in macrophages. Another neuronal repellent cue, Semaphorin 3a, induces macropinocytosis in neurons[91], which might be due to the activation of Rac1[92,93]. Additional studies are needed to completely elucidate the molecular machinery by which NSlit2 suppresses macropinocytosis and cell migration, including its actions on Rho-family small GTPases other than RhoA/Rac1/Cdc42, which are also expressed in macrophages[50].

Little is currently known about the physiological relevance of macropinocytosis by immune cells in an in vivo setting[94,95]. We

demonstrate here that not only in vitro, but also in vivo NSlit2 significantly attenuates macropinocytosis. The functional implications may be quite important given the key role of macropinocytosis in immune surveillance by macrophages and iDCs[8]. Some viruses, such as Ebolavirus[96], use the macropinocytic route to enter host cells[6,8]. In addition, constitutive macropinocytosis, a unique function of macrophages and iDCs, enables them to engulf soluble bacterial PAMPs, such as MDP, that bind to intracellular pathogen recognition receptors (PRRs), such as NOD2. In the present study, we found that NSlit2 attenuated the ability of a bacterial peptidoglycan, MDP, to be taken up by macrophages and to subsequently induce the activation of NF-κB via NOD2[5,97]. Additional work is needed to investigate whether NSlit2 has a therapeutic potential in inflammatory conditions associated with hyper-activation of NOD2 signaling[98]. It will be intriguing to see if and how SLIT2 affects the entry of ligands for other cytosolic PRRs, such as NOD1, in future studies.

Pathological stimuli such as LPS, which are associated with initial transient augmentation followed by long-term suppression of macropinocytosis in iDCs[99,100], have also been reported to reduce Slit2 mRNA levels in tissues in vivo[101]. Guan et al. reported that SLIT2 inhibits directed migration of dendritic cells in vitro and in vivo[33]. SLIT2-ROBO1 signaling also inhibits podosome formation in iDCs[102]. In addition to macrophages, iDCs also exhibit robust constitutive macropinocytosis, and this is their predominant means of antigen uptake[103]. Interestingly, endogenous levels of Slit2 mRNA are increased following antigen sensitization in vivo[33,35], but the physiological significance of this phenomenon remains to be elucidated. Future work will investigate whether SLIT2 causes long-term suppression of macropinocytosis in iDCs.

MDP signaling results in the upregulation of circulating levels of pro-inflammatory cytokines, including CXCL1, in mice[64]. Interestingly, we found that in a murine model of sterile peritonitis, both NSlit2 and a pharmacologic inhibitor of macropinocytosis, EIPA, attenuated the MDP-induced CXCL1 secretion, locally in the peritoneal exudate (Supplementary Fig. 5b), and systemically in serum (Fig. 5e, f). The effects of NSlit2 on MDP-NOD2 signaling are summarized in Supplementary Fig. 5f. We found that endogenous SLIT2 protein is concentrated in the peritoneal membrane, but SLIT3 is not (Fig. 5g, Supplementary Fig. 5c, d). Currently, no commercially available, validated ELISA kit is available to measure SLIT1 protein in murine tissues. However, Wu et al. previously reported that Slit1 mRNA is not detected in adult rodent non-neuronal tissues[29]. In line with those findings, Burgstaller et al. recently used mass spectrometry to demonstrate the presence of SLIT2 and SLIT3 proteins in the lung, but they did not detect any SLIT1[104]. We, therefore, infer that SLIT2 is the major endogenous Slit protein present locally in the peritoneal membrane.

Because SLIT2 or ROBO1 knockout mice do not survive postnatally beyond 3 weeks[71,72], we used soluble Robo1N fragment to block the effect of endogenous SLIT2 in vivo. MDP-induced secretion of CXCL1 was significantly enhanced in Robo1N-treated mice, and this effect was blocked by EIPA (Fig. 5h). Together, these data suggest that endogenous SLIT2 negatively regulates MDP signaling and the consequent secretion of inflammatory chemokines, at least in part, by inhibiting macropinocytosis. While the present study focused on acute MDP-NOD2 signaling, chronic NOD2 activation has been implicated in the development of tolerance to bacterial PAMPs in intestinal macrophages[105]. Recent studies also highlight the physiological role of NOD2 signaling in non-immune cells[106,107]. Future studies will investigate how SLIT2 affects these aspects of NOD2 signaling.

During systemic infection producing sepsis, vascular leak may occur, and SLIT2 may influence these changes in the vascular integrity. Indeed, several studies have highlighted the vaso-protective role of SLIT2, which prevents increased vascular permeability induced by pro-inflammatory stimuli in vivo[108–110]. These actions of SLIT2 were proposed to occur through the endothelial cell-specific Roundabout receptor, ROBO4[111], but a more recent study challenges this idea[112]. In addition, the role of SLIT2 in maintaining vascular integrity under basal conditions is not clear. Furthermore, the differential contributions of endothelial ROBO1 and ROBO2 receptors in mediating the actions of SLIT2 on vascular integrity and permeability remain to be elucidated[23]. A better understanding of how Slit and Robo proteins coordinate the functions of immune cells and endothelial cells is needed to reveal the interplay of these pathways in shaping immune responses associated with infection and inflammation.

Recent studies from independent groups have shed light on complex signaling pathways involved in the metabolic reprogramming in KRAS-transformed cells, which use macropinocytosis as the primary route for the uptake of large proteins, such as albumin, in a nutrient-deficient tumor microenvironment[11,12,17]. In another recent study, Göhrig et al. reported that SLIT2 expression is absent in the KRAS-transformed PDAC cell lines, MiaPaCa-1 and PANC-1. In addition, ectopic expression of SLIT2 resulted in reduced primary tumor size and secondary neural invasion in vivo[38]. To test the hypothesis that SLIT2 could inhibit macropinocytosis in cancer cells, we chose two KRAS-transformed adenocarcinoma cell lines, PANC-1 and DLD-1. Both cell lines were able to take up albumin, via macropinocytosis, when grown in Glut-deprived culture medium (Fig. 6a). This uptake was blocked by NSlit2 only in ROBO1-expressing PANC-1 cells[38], but not in DLD-1 cells, which lack ROBO1 (Fig. 6d)[69,70]. This could, in part, explain the reduced primary tumor size observed in SLIT2-expressing PDACs[38]. Based on current knowledge, ROBO1 plays either a tumor suppressor or a promoter role, depending on the tissue involved and type of cancer[36]. Further elucidation of mechanisms by which SLIT2/ROBO1 signaling simultaneously inhibits cancer cell migration and macropinocytosis will be important for potential development of new oncologic therapies.

We here provide evidence for an endogenous inhibitor of macropinocytosis. In addition, we provide insight into the physiological relevance of SLIT2 signaling in cells performing macropinocytosis to limit the entry of immunogenic microbial products, which activate cytosolic PRRs. In summary, NSlit2, acting together with ROBO1, has pleiotropic effects on recruitment, adhesion, and immune activation of monocytes and macrophages, not only though inhibition of Rac and Cdc42, but also via inactivation of MYO9B and consequent activation of RhoA. Our study challenges the notion that signals that inhibit cell migration enhance macropinocytosis, and vice versa. We also provide evidence that SLIT2 can attenuate the growth of KRAS-transformed cancer cells by inhibiting macropinocytosis. Our results suggest that NSlit2, in addition to inhibiting cancer cell migration, may also directly inhibit the macropinocytic uptake of nutrients by cancer cells, thereof inhibiting tumor growth. Further studies will explore the precise mechanisms by which SLIT2-induced signaling is spatiotemporally regulated in different cell types and diverse tissue microenvironments.

## Methods
**Reagents and antibodies**. Phalloidin dyes labeled with AF-488 or AF-594, DAPI (4′,6-diamidino-2-phenylindole dihydrochloride), Hoechst 33342 dye, and rabbit polyclonal anti-ROBO1 antibody were purchased from Thermo Fisher Scientific (Rockford, IL, USA). Total and active RhoA, and active Rac1/2/3 G-LISA kits, Rho inhibitor I (TAT-C3), Rho activator II (CN03), and mouse monoclonal anti-RhoA antibody were bought from Cytoskeleton Inc. (Denver, CO, USA). Rabbit monoclonal antibodies against total NF-κB p65 and phospho-NF-κB p65 (Ser536) were from Cell Signaling Technology (Danvers, MA, USA). Anti-CD68 and Anti-c-Myc

antibodies were bought from BioLegend (San Diego, CA, USA) and Abcam (Cambridge, MA, USA), respectively. Tetramethylrhodamine (TMR)-labeled, 70,000 MW, lysine fixable dextran was from Life Technologies (Carlsbad, CA, USA). Lympholyte-H® was bought from Cedarlane (Burlington, ON, Canada). The recombinant murine and human CSF1 were purchased from PeproTech (Rocky Hill, NJ, USA). Paraformaldehyde (PFA; 16% wt/vol) was bought from Electron Microscopy Sciences (Hatfield, PA, USA). Dako mounting medium was from Agilent technologies (Santa Clara, CA, USA). CK-666 and SMIFH2 were bought from Sigma-Aldrich Canada (Oakville, ON, Canada). Blebbistatin was from Abcam (Cambridge, MA, USA). SuperblockTM blocking buffer in TBS was from Thermo Fisher Scientific (Rockford, IL, USA). Unconjugated MDP and MDP conjugated with FITC (MDP-FITC) were purchased from InvivoGen (San Diego, CA, USA). All cell culture media, and buffer solutions, unless mentioned otherwise, were purchased from Wisent (St-Bruno, QC, Canada). ELISA kits for measuring murine CXCL1, SLIT2, and SLIT3 proteins were purchased from R&D Systems (Minneapolis, MN, USA), CUSABIO (Wuhan, China), and LSBio (Seattle, WA, USA), respectively. The DCTM protein assay kit was from Bio-Rad Laboratories (Mississauga, ON, Canada). RNeasy® Plus Mini Kit and Superscript VILO III Mastermix were purchased from Qiagen Canada (Toronto, ON, Canada) and Thermo Fisher Scientific (Rockford, IL, USA), respectively. JumpStart™ REDTaq® Ready-Mix™ Reaction Mix was bought from Sigma-Aldrich Canada (Oakville, ON, Canada). Recombinant Robo1N protein was purchased from R&D Systems (Minneapolis, MN, USA).

**Plasmids.** The pEGFP-C3-RAC1-Q61L plasmid was from S.G.'s laboratory[113]. The pEGFP-C3 and pEYFP-N1 plasmids were purchased from Takara Bio USA, Inc. (formerly Clontech, Mountain View, CA, USA). The pcDNATM3.1(+)-ROBO1 plasmid was a kind gift from Dr. Tony Pawson's group (University of Toronto, Toronto, ON, Canada). The pEYFP-N1-ROBO1 plasmid was made by digesting pcDNATM3.1(+)-ROBO1 and pEYFP-N1 plasmids with NheI and HindIII restriction enzymes and ligating ROBO1-coding sequence (CDS) into the pEYFP-N1 backbone. A plasmid with partial MYO9B CDS in pCMV-SPORT6 vector was purchased from PlasmID Repository of Dana-Farber/Harvard Cancer Center DNA Resource Core. The partial CDS encoded for C-terminal amino acid residues 1600–1780, which includes the RhoGAP domain (amino acid residues 1703–1888), of human MYO9B protein isoform 1 (accession—NP_004136.2). The CDS was put into pcDNATM3.1(+) vector (Thermo Fisher Scientific, Rockford, IL, USA) with six myc tags between EcoRI and EcoRV restriction sites. The silencing RNAs (siRNA) against mouse RhoA (sense 5′–3′ sequence- AGCCCUGAUAGUUUAGAAAtt) and mouse Robo1 (sense 5′–3′ sequence- GGGAAGAACUGUGACGUUU) were bought from Ambion® (Life Technologies, Carlsbad, CA, USA) and DharmaconTM (GE Healthcare, Chicago, IL, USA), respectively.

**Recombinant Slit proteins purification and endotoxin testing.** The full-length SLIT2 protein (human SLIT2 isoform 1 accession number: NP_004778.1) has 4 leucine-rich repeats (LRR; D1-D4), 9 EGF-like domains, a Laminin G-like (LamG) domain, and a C-terminal cysteine knot-like (CK) domain (Fig. 1b). Recombinant NSlit2 included AA 26-1118 of SLIT2. In Slit2ΔD2, the ROBO1/2-binding D2 LRR (AA 235–444) was removed from NSlit2 and replaced with a short linker[22]. CSlit2 comprised AA 1268–1525 of full-length SLIT2 protein[32]. Human SLIT2 cDNA (encoding AA 26–1529 of the protein, NP_004778.1) was amplified using forward (5′-CTATCTAGACCTCAGGCGTGCCCGGCGCAGTGC-3′) and reverse (5′-CTAGGATCCGGACACACACCTCGTACAGC-3′) primers and cloned into the pTT28 vector (a kind gift from Dr. Yves Durocher, Montreal, QC, Canada) which contains a C-terminal (His)8G tag[114] between the NheI and BamHI restriction sites. Large-scale preparation of recombinant proteins was performed by transient transfection of HEK293-EBNA1 cells as described previously[30]. Briefly, HEK293-EBNA1 cells were transfected with 1 μg/ml cDNA[115] and culture medium was collected 5 days after the transfection. The recombinant proteins were isolated using Fractogel-cobalt column purification of the conditioned medium[30]. NSlit2, CSlit2, and Slit2ΔD2 preparations were tested for endotoxin levels using a Tox-insensorTM Chromogenic LAL Endotoxin Assay Kit (GenScript, Piscataway, NJ, USA). Endotoxin levels in all preparations were less than 0.05 EU/ml and are presented in Supplementary Table 1.

**Primary monocyte/macrophage isolation and cell culture.** For BMDM isolation, 8–12-week-old mice were euthanized using the $CO_2$ inhalation method and femurs and tibias were cleaned and excised. The bone marrow cells were pelleted and resuspended in 1 ml of phosphate-buffered saline (PBS) and filtered through 100 μm nylon mesh. Filtered cells were centrifuged at 5500 g for 5 min and cultured in RPMI 1640 medium supplemented with 10% heat-inactivated fetal bovine serum (FBS), 25 ng/ml murine CSF1 and 1% penicillin/streptomycin/amphotericin B for 7–10 days.

The protocol for human participation and blood donation (#1000060065) was reviewed and approved by The Hospital for Sick Children Research Ethics Board, Toronto, ON, Canada. Written, informed consent was obtained from all participants before the blood donation. Peripheral blood mononuclear cells from 60 ml of healthy donor blood were separated using Lympholyte-H. The cells were plated and grown in two 10-cm dishes in RPMI 1640 medium supplemented with

10% heat-inactivated FBS and 25 ng/ml of human CSF1 and 1% penicillin/streptomycin/amphotericin B for 7–10 days.

RAW264.7, HEK293T, DLD-1, and PANC-1 cell lines were purchased from the American Type Culture Collection (ATCC, Manassas, Virginia, USA) and cultured in Dulbecco's Modified Eagle Medium (DMEM) medium supplemented with 10% heat-inactivated FBS and 1% penicillin/streptomycin/amphotericin B (complete DMEM). HEK293T-ROBO1 (HEK293T cells stably expressing human ROBO1) cells were grown in complete DMEM supplemented with 1000 μg/ml of G418. HEK293-EBNA1-6E cells were cultured in FreestyleTM F17 culture medium from Thermo Fisher Scientific (Rockford, IL, USA).

**Cell transfections.** RAW264.7 cells were transiently transfected with either pEGFP-C3-RAC1-Q61L or the control pEGFP-C3 plasmid (Takara Bio USA Inc., CA, USA) using the AmaxaTM Nucleofactor II system (Lonza, Basel, Switzerland). Transfections were done using the manufacturer's recommended protocol (D32 setting) and the experiments were performed 48 h (h) after transfection. The MYO9B-GAP-Flag or Flag plasmids were transfected in RAW264.7 cells using Viromer® Red transfection reagent (Lipocalyx GmbH, Halle, Germany), per the manufacturer's instructions. Cells were used for spreading assays 48 h after transfection. The siRNA transfections were done using HiPerFect® transfection reagent (Qiagen, Hilden, Germany) in six-well plates as per the manufacturer's instructions (final siRNA concentration- 50 nM) for 72 h.

**Immunoblotting.** Cells were washed with ice-cold PBS thrice and lysed using 1× RIPA (Abcam) buffer (500 μl per well for a 6-well plate) supplemented with protease inhibitor cocktail (Sigma-Aldrich Canada, Oakville, ON). In the case of the phospho-protein blotting, 1× phosphatase inhibitor (Sigma-Aldrich Canada, Oakville, ON) was added to the lysis buffer. Protein concentration was determined using Bio-Rad DC protein assay (Bio-Rad laboratories, Mississauga, ON, Canada). Fifty micrograms of protein per sample was loaded and proteins were separated by SDS-PAGE. The proteins were transferred to a PVDF membrane and blocked in 5% fat-free milk in Tris-buffered saline (TBS) containing 0.05% Tween-20 for 1 h. For phospho-proteins, SuperBlockTM in TBS was used instead of milk. All primary antibodies were incubated as indicated in Supplementary Table 2. The membrane was washed three times with TBST for 10 min each time and HRP-conjugated secondary antibodies (1:5000) were added for 1 h at room temperature. The membrane was washed with TBST, treated with SupersignalTM Chemiluminescent substrate (Thermo Fisher Scientific, Rockford, IL, USA) and visualized on a ChemiDoc MP imaging system (Bio-Rad laboratories, Mississauga, ON, Canada). Band intensity was quantified using ImageJ software, version 1.51v. Following phospho-protein blotting, the same membrane was stripped using RestoreTM western blot stripping buffer (Thermo Fisher Scientific, Rockford, IL, USA) and probed for total protein levels. Detailed information for all primary antibodies used in this study is provided in Supplementary Table 2.

**RNA isolation, reverse transcription (RT) and polymerase chain reaction (PCR).** Total RNA was isolated from cells and tissues using RNeasy® Plus Mini Kit from Qiagen and was stored at −80 °C until use. RT was performed using Superscript VILO Mastermix with following conditions: 25 °C 10 min, 42 °C 1 h, 85 °C 5 min and products were stored at −20 °C until PCR. The PCR was performed with JumpStart REDTaq reaction mix using the following cycling conditions: initial activation at 94 °C 2 min, followed by 35 cycles of 94 °C (30 s), 57 °C (30 s), 72 °C (2 min) and final extension at 72 °C for 5 min. PCR products were run on 2% agarose gel.

All primer sequences for PCR are provided in Supplementary Table 3.

**In vitro recombinant Slit2 and other pharmacological inhibitor treatments.** All treatments were performed in independent triplicates. Cells were serum-starved for 1 h before all treatments. For Slit treatments, cells were suspended using Accutase® (Stemcell technologies, San Diego, CA) and incubated with 30 nM of purified human NSlit2, CSlit2, and Slit2ΔD2, respectively, in serum- and antibiotic-free RPMI 1640 for 15 min at 37 °C in the suspended state. Cells ($5 \times 10^4$ cells per well) were plated on 12-well coverslips pre-coated with poly-D-lysine, incubated for 1 h at 37 °C and then fixed with 4% PFA and permeabilized with Triton X-100 (0.1%) for 10 min each. Cells were incubated with phalloidin labeled with AF-488 (or AF-594 when used for cells expressing GFP or YFP-tagged plasmids) and DAPI and coverslips were mounted on 1-mm-thick glass slides. For Robo1N in vitro experiments, 30 nM NSlit2 was pre-incubated with 90 nM Robo1N (molar ratio-1:3) for 1 h before adding to the cells. For Rho Activator II and TAT-C3 treatments, cells were incubated with the reagents (2 μg/ml) for 4 h before Slit treatments. For all other pharmacologic inhibitors, cells were pre-treated before adding recombinant Slit as follows: Blebbistatin (50 μM, 30 min), SMIFH2 (10 μM, 30 min), CK-666 (100 μM, 60 min), Y-27632 (10 μM, 30 min).

**Confocal microscopy.** All confocal images were acquired using a spinning disk confocal microscope (Leica DMi8) equipped with a Hamamatsu C9100-13 EM-CCD camera and 63× (NA- 1.4) or 40× (NA- 1.3) oil immersion objectives. Cell surface area and cell rounding (shape factor) were measured using Volocity 6.3 software (PerkinElmer, Waltham, MA, USA).

Shape factor is a measure of how similar a three-dimensional shape is to a perfect sphere. It is defined as the ratio of the surface area of a sphere with the same volume as the given object (in this case, a cell) to the surface area of the object. The shape factor is 1 for a perfect sphere; becoming smaller for more irregular shapes.

**RhoA activation, total RhoA, and Rac1/2/3 activation G-LISA assays**. Raw 264.7 cells were serum starved for 4 h prior to the assay. Cells were then treated with 30 nM of purified NSlit2, CSlit2, or Slit2ΔD2 in serum-free RPMI 1640 medium for 15 min at 37 °C. Cells were lysed with protein lysis buffer containing the protease inhibitor mixture provided in the kits. The cleared protein lysate supernatants were normalized to the same amount of protein input using the protein detection reagent included in the kit. Activated RhoA, activated Rac1/2/3, and total RhoA assays were performed according to the manufacturer's instruction (Cytoskeleton, Denver, CO). The results were read at 490 nm by a Molecular Devices Visible Light Plate Reader (Molecular Devices VersaMax 190).

**In vitro macropinocytosis**. Cells were gently lifted using accutase and plated on 18 mm coverslips in a 12-well plate 24 h prior to the assay. All solutions were prepared in serum-free, phenol red-free RPMI medium containing calcium. Following Slit treatments (30 nM) for 15 min at 37 °C, TMR-conjugated 70 kDa dextran (100 μg/ml) or albumin-FITC (Fig. 6b, c, 500 μg/ml) was added to the media and incubated for 15 min at 37 °C. Cells were washed with PBS (5 min), trypsin (20 s) and PBS (5 min) to remove all unbound TMR-dextran and fixed with 4% PFA and stained for nuclei with Hoechst dye. The micrographs were acquired using a spinning disk confocal microscope (Leica DMi8) equipped with Hamamatsu C9100-13 EM-CCD camera and 63× (NA- 1.4) objective. Macropinosomes were quantified using the measurement tool in ImageJ software, version 1.51v from areas of two-dimensional particles from projections of three-dimensional Z-stacks[5,116]. Macropinosomes were defined as TMR-dextran-positive intracellular vesicles larger than 200 nm in diameter[5,8].

**Cancer cell Glut deprivation viability assay**. DLD-1 and PANC-1 cells were seeded ($10^4$ cells per well) in a 96-well plate and grown in DMEM without L-Glut supplemented with 0.5 mM Glut (physiological levels), 0.2 mM Glut (low Glut condition), and 2% albumin, when indicated, for 6 days. The macropinocytosis inhibitor, EIPA (25 μM), was added to the medium for the last 2 days only, where indicated. In some experiments, 30 nM NSlit2 or Slit2Δ2 was added and the medium was changed every 24 h, for 6 days. At the end of all treatments, cell viability was assessed using a MTT assay kit (Roche Diagnostics purchased from Sigma-Aldrich, Oakville, ON, Canada).

**In vitro MDP-FITC treatment**. Primary murine BMDM were suspended using accutase and attached to uncoated 18 mm coverslips 24 h prior to the treatment. BMDM were serum-starved for 1 h and treated with either serum-free medium (control) or NSlit2, CSlit2, or Slit2ΔD2 (30 nM each) at 37 °C for 15 min. MDP-FITC (1 μg/ml) was added to the medium and cells were incubated at 37 °C for an additional 30 min. The cells were fixed using 4% PFA and prepared for confocal imaging as described earlier in the 'In vitro macropinocytosis' section. For immunoblotting experiments, BMDM were grown in six-well plates. Following the Slit and MDP treatments as described above, cells were lysed using RIPA buffer 5 min after the MDP treatment.

**Animal work**. In vivo macropinocytosis and MDP treatment protocols (Animal User Protocol—44731) were reviewed and approved by the Animal Care Committee of The Hospital for Sick Children (Toronto, ON, Canada) in accordance with guidelines established by the Canadian Council on Animal Care (CCAC). The generation of Rac1/2 double KO (DKO) mice has been described previously[117,118] and was carried out by M.G.'s lab in accordance with the Guide for the Humane Use and Care of Laboratory Animals and approved by the University of Toronto Animal Care Committee. Eight-to-twelve-week-old mice were used for all in vivo experiments.

Sample sizes (n) for all in vivo experiments were calculated using G*Power software version 3.1.9.2 (Universität Düsseldorf, Germany)[119].

**In vivo macropinocytosis**. The murine model of sterile peritonitis has been described previously[63]. Six animals were used for each of the four treatment groups. On day-0, C57BL/6J mice were treated with sodium periodate (200 μg in 200 μl) by a single intraperitoneal (i.p.) injection to induce sterile peritonitis. Forty-eight hours after the periodate injection, animals were injected with either saline, or equivalent molar amounts of recombinant Slit2 proteins, namely, NSlit2 (1 μg), CSlit2 (0.2 μg), or Slit2ΔD2 (1 μg) by an intravenous (i.v.) route. On day 3, animals were injected i.p. with TMR-labeled 70 kDa dextran (200 μg) and euthanized after 30 min (Fig. 4g). The peritoneal macrophages were isolated as described by Ray et al.[120] and centrifuged at 300 × g for 10 min. The cell pellet was resuspended in serum-free RPMI 1640 and allowed to attach to poly-D-lysine-coated coverslips for 1 h. The coverslips were washed with PBS twice and cells were fixed with 4% PFA

and blocked using 5% goat serum. The coverslips were incubated with anti-CD68 antibody, a pan monocyte/macrophage marker, conjugated with AF-488 (BioLegend, San Diego, CA, USA) in 5% goat serum overnight at 4 °C. The coverslips were washed with PBS three times and incubated with Hoechst dye in the blocking solution for 1 h at 4 °C. The coverslips were washed with PBS three times and mounted using Dako mounting medium.

**In vivo MDP treatment**. Four animals were used for each treatment group. On day-0, C57BL/6J were treated with sodium periodate (200 μg in 200 μl) by a single i. p. injection to induce sterile peritonitis. Seventy-two hours after the periodate injection, saline vehicle, or NSlit2 (1 μg), or EIPA (25 mg/kg body weight in 200 μl) was administered i.p. After 1 h, MDP (10 ng) was administered i.p. After 2 h, blood was collected by intracardiac puncture under isoflurane anesthesia. Mice were euthanized immediately following the blood collection, and peritoneal exudates were collected by lavaging the peritoneal cavity with 5 ml cold PBS. Cells in the peritoneal exudate were counted using a hemocytometer. For serum collection, blood was allowed to clot for 2 h at room temperature and centrifuged at 1000 × g for 15 min. Serum aliquots were stored at −20 °C until the assay. Serum CXCL1 levels were determined by ELISA. For experiments involving in vivo Robo1N treatment, 72 h after sodium periodate treatment, saline vehicle, Robo1N (7 μg) or EIPA (25 mg/kg body weight), and Robo1N together were administered i.p. One hour later, MDP (10 ng) was administered i.p. After 2 h, blood was collected by intracardiac puncture. Mice were euthanized immediately, and the peritoneal exudates were collected. Murine CXCL1 ELISA was performed using a microplate reader as per the manufacturer's recommendations (R&D Systems, DY453).

**Peritoneal membrane isolation**. Serum samples were collected from C57BL/6J mice (ten males and ten females) as described before and animals were euthanized immediately after the blood collection. Under aseptic conditions, the abdominal cavity was opened by removing the skin and underlying muscle but keeping the peritoneal membrane intact. The peritoneal membrane was separated from underlying adipose tissue and organs and collected in cold PBS. The membrane was rinsed with PBS once and resuspended in RIPA buffer. Samples were homogenized with a tissue homogenizer and stored at −20 °C overnight. Two freeze-thaw cycles were performed to break cell membranes and homogenates were centrifuged at 5000 × g for 5 min at 4 °C. Supernatants were stored at −80 °C.

**Murine SLIT2 and SLIT3 ELISA**. SLIT2 and SLIT3 ELISA were performed using a microplate reader as per the manufacturer's recommendations (CUSABIO CSB-E11039m and LSBio LS-F-7173, respectively). Briefly, 100 μl samples and standards were incubated in pre-coated assay wells at 37 °C for 2 h. Samples were removed and wells were then incubated with biotin antibody at 37 °C for 1 h, with avidin antibody for 1 h, and TMB substrate for 30 min. Wells were thoroughly washed three times between incubation steps. After incubation with the TMB substrate, 50 μl stop solution was added to each well and the optical density was read immediately. Wavelength correction was applied by subtracting readings at 540 nm from those at 450 nm.

**Statistical analyses**. All data are presented as mean ± standard error of mean (SEM). Statistical tests were performed using GraphPad Prism 7 software (San Diego, CA, USA) and are described in the corresponding figure legends. $p < 0.05$ was considered statistically significant. All p values are reported up to four decimal places.

**Biological materials**. Unique biological materials used in this study will be made available upon a reasonable request from corresponding author (L.A.R.).

**Reporting summary**. Further information on research design is available in the Nature Research Reporting Summary linked to this article.

## Data availability

All data supporting the findings of this study are available within the paper, its supplementary information files, and source data are provided as a Source Data file linked to this article. Source data underlying Figs. 1(a, d–h), 2(a, b, d–f), 3(b, c, e, f), 4(b, c, e, f, i), 5(a–h), 6(a–d), and Supplementary Figs. 1(a, c–e), 2(a–e), 3(a–d, f, g), 4(b–d, f), 5(a–c, e) are provided as a Source Data file linked to this article. Source data are provided with this paper.

## Code availability

This study did not generate any new code. Source data are provided with this paper.

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

## Acknowledgements

We thank Paul Paroutis and Kimberly Lau for their assistance in imaging analysis. We thank Dr. Spencer Freeman for scientific advice and discussion. We thank Dr. Marvin Estrada for his technical assistance for in vivo work. This work was supported by grants from Canadian Institutes of Health Research (CIHR) to L.A.R. (MOP111083 and MOP136896). B.W.P. was supported by Queen Elizabeth II/Heart and Stroke Foundation of Ontario (HSFO) Graduate Scholarship in Science and Technology. J.Y.W. is supported by National Institutes of Health grants (RO1CA175360 and RO1NS107396). D.J.P. is supported by CIHR grant, FDN-14333.

## Author contributions

Conceptualization; V.K.B., S.G., L.A.R.; Methodology; V.K.B., T.M., Y.W.H.; Investigation; V.K.B., T.M., Y.W.H., S.P., B.W.P., G.Y.L.; Writing—Original Draft: V.K.B.; Writing—Review & Editing: V.K.B., L.A.R., T.M., D.J.P., S.G.; Resources: L.A.R., D.J.P., S.G., M.G., J.Y.W.; Funding acquisition: L.A.R.; Supervision: L.A.R. All authors reviewed and approved the final submission.

## Competing interests

The authors declare no competing interests.
