## [Peer Review File · Nature Communications]

Reviewers' comments:

Reviewer #1 (Remarks to the Author):

In this manuscript by Bhosle and colleagues, they follow on from their previous work on Slit2 signalling in immune cells to demonstrate that it is also able to suppress macropinocytosis in macrophages. Importantly they also show a potentially important physiological role for this as they demonstrate that Slit2-mediated macropinocytotic suppression is able to affect subsequent inflammasome signaling (or NF κ B phosphorylation at least).

Describing a novel pathway for both the regulation of macropinocytosis and subsequent inflammatory signaling, this is an interesting and potentially important study. However some of their proposed molecular mechanism underpinning the cytoskeletal regulation by Slit2 is weakly supported in places and poorly controlled. There are therefore several major experimental concerns that need to be addressed in order to support their conclusions.

Specific points:

The mechanistic experiments indicating that Slit2 functions via activation of RhoA largely only use cell spreading as the functional readout. They then use this data to argue that the suppression of macropinocytosis acts through the same pathway, rather than regulation of Rac1 as previously reported. Whilst in supplementary figure 4B they show that RhoA siRNA is able to partially suppress the inhibition of macropinocytosis by Slit2, as this also appears to increase the basal level of fluid uptake, and there is still a significant effect of Slit2, this data on its own is unconvincing.

Whilst the proposed Slit2/Robo/RhoA pathway may be true for cell spreading, given that they, and others have previously proposed a mechanism for inhibition of migration via Rac1 they should also confirm how Rac1 activity changes in these experiments. As currently written, it is suggested that the RhoA and Rac1 pathways are mutually exclusive. There is no reason to assume this. It seems more likely that both pathways are manipulated by Slit2, and it would be consistent with the literature if perhaps spreading is more sensitive to RhoA activity, whilst migration and macropinocytosis are more dependent on Rac1. Indeed they show that manipulation of Rac1 does not effect spreading in Figure 2A, whereas the role of Rac in macropinocytosis and migration is well-established. They should show whether both Rac1 and RhoA are altered upon Slit2 signaling in their experiments, and test the relative contribution of each to the observed effects on macropinocytosis at least, and ideally migration (better, but optional).

In Figure 5 they show that the rate of macropinocytosis correlates with inflammasome signaling, indicated by p-NF- κ B levels. However this does not prove causation and as they state in their discussion may be due to other entry routes of MDP. This should be confirmed by manipulating macropinocytosis in these experiments.

In their siRNA, but only show data using a single oligo. This is open to artefacts and insufficiently rigorous. This should be confirmed either using multiple independent oligos or rescue experiments.

Minor points:

The pathway they describe is sufficiently complex that it would help the reader to include a model figure showing each protein, and the target of each inhibitor they use.

They use the term cell surface area to describe how much cells have spread throughout the results. This is inaccurate, as they are measuring cross-sectional area, or spreading extent rather than the actual cell surface. This should be corrected.

Sometimes they refer to Myo9B and others Myo9B-GAP. I presume they only overexpress the GAP

domain, but should check this is accurate throughout.

Figure 2D X-legend states dD2Slit2, whereas other panels refer to Slit2dD2. Be consistent.

Reviewer #2 (Remarks to the Author):

In this study, Bhosle and colleagues provide evidence suggesting that the secreted protein Slit2 promotes macropinocytosis in macrophages via Robo1 and RhoA. Almost all the work is conducted in vivo using primarily the RAW264.7 mouse cell line. Overall the data are not very original as the influence of Slits on immune cells has been extensively studied. It is already known that Slit activates RhoA and myosin in monocytes and other cell types. Most of the figures are also simple "old-fashioned" histograms and there is basically no in vivo data supporting the model. There are many mouse models available (Slit and Robo knockouts) that could have been used to support the model.

The authors only used Slit2 and they have not tested the other Slits (Slit1 and Slit3) which also bind to Robo1. What supports the specificity of Slit2 activity in this model? They do not provide any in vivo data on the expression of these molecules in inflammatory models.

Importantly, a previous study (not cited) showed that Slit3 influences monocyte migration and the effect on cell spreading appear similar.

Geutskens et al., J Immunol. 2010 Dec 15;185(12):7691-8.

The chemorepellent Slit3 promotes monocyte migration.

Likewise, why do they assume that Robo1 is the only receptor involved? Does the cell lines used (RAW264.7 and HEK293T) and human macrophage and BMDMs only express this Slit receptor?

The authors failed/neglected to cite many papers (sometimes in other cellular models) that already provided similar data (and often used the same drugs or blocking agents, such as blebbistatin or Y-27632), in particular on RhoA. This is not acceptable.

Wang et al., Glia. 2013 May;61(5):710-23. Repulsive migration of Schwann cells induced by Slit-2 through Ca²⁺-dependent RhoA-myosin signaling.

Liu et al. J Biol Chem. 2012 May 18;287(21):17503-16.

Slit2 regulates the dispersal of oligodendrocyte precursor cells via Fyn/RhoA signaling.

Backer et al., Development. 2018 Oct 2;145(19). Trio GEF mediates RhoA activation downstream of Slit2 and coordinates telencephalic wiring.

Geutskens et al., J Immunol. 2010 Dec 15;185(12):7691-8.

The chemorepellent Slit3 promotes monocyte migration.

Guan et al., Cell. 2007 Apr 20;129(2):385-95.

Long-range Ca²⁺ signaling from growth cone to soma mediates reversal of neuronal migration induced by slit-2.

What is the "shape factor"? this is rather unclear from the results and methods and they should not expect the readers to look at the Volocity manual to find out what it is.

As said before many related papers are completely ignored

Zhao et al., J Immunol. 2014 Jan 1;192(1):385-93. Slit2-Robo4 pathway modulates lipopolysaccharide-induced endothelial inflammation and its expression is dysregulated during endotoxemia.

This paper show that Slit2 represses inflammatory responses by inhibiting the Pyk2-NF- κ B pathway downstream of LPS-TLR4.

For several experimental conditions, there are alternative explanation or indirect effect that are not considered.

Reviewer #3 (Remarks to the Author):

Bhosle et al

I felt this manuscript delivered significantly less than was promised in the title and abstract. Macropinocytosis has emerged as an important endocytic pathway in various cell types and while activators have been described (growth factors, TLR ligands....) the authors are broadly correct in stating that no endogenous/physiological inhibitors are known. The authors claim that the Slit2 ligand upstream of Robo1 signalling is such a physiological inhibitor in macrophages. What is actually shown is that injection of a recombinant fragment of Slit2 (NSlit2) reduces constitutive endocytosis by peritoneal macrophages. Macropinocytosis is inferred by the large size of the endosomes observed. However, the claim that Slit2/Robo1 signalling is a physiological inhibitor of macropinocytosis is not exemplified by a physiological scenario where macropinocytosis in vivo is regulated by endogenous Slit2 as opposed to exogenous Slit2.

Main points

In spite of the focus on macropinocytosis in the title and abstract, most of the paper deals with a macrophage spreading/rounding phenotype induced by the Slit2 fragment. This aspect is hardly referred to in the Abstract. Macropinocytosis per se is only studied in 1 of the 5 main Figures (Figure 4), — 2 if you include Figure 5 which relates to NOD signalling following putative macropinocytic uptake of the MDP ligand. The Abstract does not accurately reflect the data presented in the study.

Nslit2 triggers a cell rounding phenotype (Figure 1) via RhoA activation (Figure 2). It is good to see some mechanistic basis for these effects of active Rho on cell spreading – the authors find evidence for RhoA driven formin activation (though the inhibitor used, SMIFH2 has also been reported to have other effects e.g. on p53) but how is RhoA activated by NSlit2? The authors state in the Discussion that inhibition of Myo9b (contains a RhoGAP) is the mechanism of activation of RhoA. Really? What is shown (in Figure 3) is vector-driven expression of Myo9b GAP and its effects on Nslit2/Robo signalling. I could see no analysis of endogenous Myo9b and its GAP activity. Surely this is essential to make any claim about how RhoA is activated in this setting? This is important because a mechanistic basis for Slit2-mediated activation of RhoA was already addressed in a paper last year by Backer et al in Development (<http://dev.biologists.org/content/145/19/dev153692.long>). Those authors showed that Trio was the GEF responsible for RhoA activation downstream of Slit/Robo signalling. Could this be relevant

in macrophages? The authors do not address this or refer to the Backer study. Generally I found the section describing the experiments with Myo9b somewhat confusing and I may have wrongly got the impression that RhoGAP Myo9B has opposite effects in RAW and HEK293 cells: NSlit2-induced rounding was inhibited in RAW cells (as expected from the previous RhoA activation data) by Myo9b but in Robo1 transfected HEK293 expressing Myo9B GAP domain rounding was increased? Is that right? I apologise if I have missed something but this section generally was unclear and did not seem to support the notion that Slit/Robo signalling regulates endogenous Myo9b GAP activity.

The authors appear to rule out Rho-driven effects on Rac activation state as a mechanism to explain the cell rounding phenotype but do not investigate whether suppression of Rac1/2 activation might nonetheless explain the NSlit2 effect on macropinocytosis. There is a well described reciprocal antagonism of Rac and Rho activity in cells so it is quite possible (notwithstanding the result obtained in the cell rounding experiments) that RhoA activation induced by the Slit ligand shuts down Rac activity and this is the reason for the suppression of macropinocytosis. This was not investigated.

Are we looking exclusively at macropinocytosis in the in vitro and in vivo experiments described? We are reliant on the size of the endosomes shown to conclude that these are macropinosomes. The authors do not show the background dextran uptake in the absence of Csf1, the inducer of macropinocytosis used. A helpful experiment would be to show Csf1 stimulation of macropinocytosis and suppression by a well established inhibitor (e.g amiloride) alongside Nslit2 in the in vitro experiments.

Other points:

The text is sometimes hard to follow with the flow being interrupted with bracketed text containing P values and 'this condition versus that condition' explanations. The paper would be easier to read if all this had been in the Figure legends and not the text.

None of the pages or Figures are numbered.

There are no independent size markers in Fig 1A. Is the difference in human v mouse Robo1 migration due to different sized proteins?

Why was full length Slit2 never tested?

There are no controls for siRNA knockdown of Rac and Rho. To exclude off-target effects it is generally considered essential to show 2 different siRNAs and to 'rescue' the phenotype by transfection of an siRNA resistant construct for the target gene. Neither of these controls are shown.

Expression of active Rac is shown not to rescue the rounding phenotype induced by Nslit2 (Figure 2A). No data are presented showing how much Q61L Rac was expressed (relative to endogenous Rac) and in what proportion of cells. Also this expt is only meaningful if, in control cells (i.e non NSlit2 treated macrophages), Q67L Rac1 is shown to drive a response.

There are large differences in cell surface area between some expts? E.g Fig 1C 1200 μ 2 to 600 reduction on Nslit2 exposure whereas in the expt where the Arp2/3 inhibitor is tested (Supplementary 2B) the numbers are about 350 to 120? Are the cells really 4 times larger/smaller between expts?

Overall, for the reasons stated above I felt that there was not a strong and clear enough message for a high ranked journal. The claim that Slit2/Robo1 signalling is a physiological inhibitor of

macropinocytosis is not supported by a physiological scenario where this signalling pathway operates. For example, constitutive macropinocytosis in dendritic cells is progressively shut down upon physiological maturation as shown by many labs. Might that be mediated by Slit2/Robo signalling? Since macropinocytosis is often an acutely stimulated process it is debateable whether physiological inhibitors are actually needed: attenuation may be simply due to the removal of the stimulus or feedback inhibition of the activation pathway. The effects of Slit2/Robo signalling on macrophages described here will nonetheless be of interest to a more specialised audience, for example in a dedicated cell biology journal.

We thank all reviewers for their constructive comments and for allowing us to address the concerns noted. As follows is a point-by-point response to each of the reviewer's comments and accompanying changes made to the figures and text.

Reviewer #1 (Remarks to the Author):

In this manuscript by Bhosle and colleagues, they follow on from their previous work on Slit2 signalling in immune cells to demonstrate that it is also able to suppress macropinocytosis in macrophages. Importantly they also show a potentially important physiological role for this as they demonstrate that Slit2-mediated macropinocytotic suppression is able to affect subsequent inflammasome signaling (or NF-kb phosphorylation at least).

Describing a novel pathway for both the regulation of macropinocytosis and subsequent inflammatory signaling, this is an interesting and potentially important study. However some of their proposed molecular mechanism underpinning the cytoskeletal regulation by Slit2 is weakly supported in places and poorly controlled. There are therefore several major experimental concerns that need to be addressed in order to support their conclusions.

Specific points:

1. The mechanistic experiments indicating that Slit2 functions via activation of RhoA largely only use cell spreading as the functional readout. They then use this data to argue that the suppression of macropinocytosis acts through the same pathway, rather than regulation of Rac1 as previously reported. Whilst in supplementary figure 4B they show that RhoA siRNA is able to partially suppress the inhibition of macropinocytosis by Slit2, as this also appears to increase the basal level of fluid uptake, and there is still a significant effect of Slit2, this data on its own is unconvincing.

We thank the reviewer for raising this very important point. RhoA is the most abundant Rho family small GTPase expressed in monocyte/macrophages (van Helden et al., 2012). In order to determine the relative contribution of Rac and RhoA to NSlit2-mediated inhibition of macropinocytosis, we performed a series of additional experiments. First, we demonstrated that macrophages express SRGAP2 (Supplementary Fig. 4D), a Rac1-specific GTPase activating protein (GAP) (Mason et al., 2011), and accordingly, that exposure of macrophages to NSlit2 inhibits Rac activity, measured using a Rac G-LISA assay (Supplementary Fig. 4C). These results are in keeping with previously reported work from an independent group (Kanellis et al., 2004). Next, we determined the effects of NSlit2 on Rac activity in the presence of a cell-permeable RhoA/B/C inhibitor, TAT-C3. We found that exposure to NSlit2 inactivates Rac, even in the presence of TAT-C3, suggesting that Rho activation and Rac inhibition are mediated by different downstream effectors of NSlit2-ROBO1. These new data are presented in Supplementary Fig. 4C of the revised manuscript. We agree with the reviewer that NSlit2-induced inhibition of macropinocytosis is blunted by RhoA knockdown but is not completely reversed. This suggests that NSlit2's effect on macropinocytosis is likely mediated by multiple mechanisms, including inactivation of Rac1. We have now clarified this point in the revised manuscript (please see pages- 10 and 17 of the revised manuscript).

2. Whilst the proposed Slit2/Robo/RhoA pathway may be true for cell spreading, given that they, and others have previously proposed a mechanism for inhibition of migration via Rac1 they should also confirm how Rac1 activity changes in these experiments. As currently

written, it is suggested that the RhoA and Rac1 pathways are mutually exclusive. There is no reason to assume this. It seems more likely that both pathways are manipulated by Slit2, and it would be consistent with the literature if perhaps spreading is more sensitive to RhoA activity, whilst migration and macropinocytosis are more dependent on Rac1. Indeed they show that manipulation of Rac1 does not affect spreading in Figure 2A, whereas the role of Rac in macropinocytosis and migration is well-established. They should show whether both Rac1 and RhoA are altered upon Slit2 signaling in their experiments, and test the relative contribution of each to the observed effects on macropinocytosis at least, and ideally migration (better, but optional).

As recommended by the reviewer, we have now determined whether Rac1 and RhoA activities are altered in our experiments. As described in our response to Comment #1 above, we observed that exposure to NSlit2 results in inactivation of Rac, even in the presence of Rho inhibition, suggesting that Rho activation and Rac inhibition are mediated by different downstream effectors of NSlit2-ROBO1. Accordingly, we further demonstrated that SRGAP2, a Rac1-specific GAP, is expressed in RAW264.7 macrophages and in primary murine bone marrow-derived macrophages (BMDM). Our results are in line with a previous report using RAW264.7 cells (Mason et al., 2011). These new data are presented in Supplementary Figs 4C and 4D and on page- 10 of the revised manuscript.

3. In Figure 5 they show that the rate of macropinocytosis correlates with inflammasome signaling, indicated by p-NF- κ B levels. However, this does not prove causation and as they state in their discussion may be due to other entry routes of MDP. This should be confirmed by manipulating macropinocytosis in these experiments.

We thank the reviewer for this important comment. Canton J. *et al.* previously demonstrated that constitutive macropinocytosis is the main route of entry of NOD2 ligands, including muramyl dipeptide (MDP), into primary human macrophages (Canton et al., 2016). However, we agree that other routes of fluid uptake (e.g. endocytosis) might promote entry of MDP into macrophages, to a lesser extent. While the binding of bacterial PAMPs, such as MDP, to intracellular NOD receptors robustly activates NF- κ B, a second signal is typically required for complete inflammasome activation (Sutterwala et al., 2014). We chose MDP as an experimental tool in this study because it selectively binds to NOD2 (and not intracellular NOD1 or other cell-surface Toll-like receptors) and is known to enter macrophages via macropinocytosis (Canton et al., 2016). To verify the role of macropinocytosis in promoting MDP-induced cellular signaling events, and to study the importance of this event *in vivo*, we used a murine model of sterile peritonitis (Mukovozov et al., 2015), in which uptake of MDP by peritoneal macrophages results in the production of the chemokine, CXCL1, which is detected in the systemic circulation (Magalhaes et al., 2008). We tested the ability of NSlit2 and the amiloride derivative, EIPA (a well-known selective inhibitor of macropinocytosis), to block the MDP-induced elevation in serum levels of CXCL1. We found that following exposure of macrophages to MDP *in vivo*, both NSlit2 and EIPA significantly decreased levels of CXCL1 not only in serum but also locally in the peritoneal fluid. We found that NSlit2 and EIPA inhibit NOD2 signaling in macrophages to a comparable extent *in vivo*. These new data are presented in the extended Fig. 5 and Supplementary Fig. 5, and on pages- 11-13 and 17-18 of the revised manuscript.

4. In their siRNA, experiments, they show data using a single oligo. This is open to artefacts and insufficiently rigorous. This should be confirmed either using multiple independent oligos or rescue experiments.

The reviewer raises an important point. To confirm the results obtained using Robo1 siRNA, we used an alternative, complementary approach, namely, we co-incubated NSlit2 with a soluble N-terminal fragment of the human ROBO1 receptor (Robo1N). Robo1N contains the NSlit2-binding Ig1 motif (Nguyen Ba-Charvet et al., 2001), and has previously been shown to act as a SLIT2 antagonist, by blocking the binding of NSlit2 to ROBO1 (Patel et al., 2012; Werbowetski-Ogilvie et al., 2006). We observed that while Robo1N alone had no effect on cell spreading or rounding, the pre-incubation of NSlit2 with Robo1N prevented the NSlit2-induced decrease in spreading and increase in rounding of macrophages. These new results are presented in Supplementary Fig. 1F-G, and on page- 6 of the revised manuscript.

Minor points:

1. The pathway they describe is sufficiently complex that it would help the reader to include a model figure showing each protein, and the target of each inhibitor they use.

We thank the reviewer for this suggestion. Accordingly, we have now included a model figure summarizing SLIT2-ROBO1 signaling and its effect on MDP-NOD2 signaling in macrophages (Please see Supplementary Fig. 5D in the revised version of the manuscript).

2. They use the term cell surface area to describe how much cells have spread throughout the results. This is inaccurate, as they are measuring cross-sectional area, or spreading extent rather than the actual cell surface. This should be corrected.

We apologize for the confusion. Indeed, we did not measure two-dimensional cell spreading, but rather, we measured the total cell surface area using all Z-stacks and integrating individual measurements using Volocity 6.3 software. We have clarified this point in the revised version of the manuscript (please see page- 5 of the revised manuscript).

3. Sometimes they refer to Myo9B and others Myo9B-GAP. I presume they only overexpress the GAP domain, but should check this is accurate throughout.

As noted by the reviewer, we used a plasmid encoding the C-terminal region of MYO9B, including the complete Rho-GAP domain (MYO9B-GAP). MYO9B-GAP has been previously shown to possess RhoA GAP activity in cells (Kong et al., 2015). Due to the challenges in transfecting macrophages with a plasmid encoding a protein as large as full-length MYO9B (the human MYO9B isoform I contains 2157 amino acids), we instead used MYO9B-GAP (Please see Supplementary Fig. 3E for sequence alignment). For the sake of clarity, we have now consistently referred to MYO9B-GAP throughout the revised version of the manuscript.

4. Figure 2D X-legend states dD2Slit2, whereas other panels refer to Slit2dD2. Be consistent.

We thank the reviewer and apologize for this oversight. For the sake of consistency, we now corrected the term Slit2ΔD2 in the legend for Fig. 2D.

Reviewer #2 (Remarks to the Author):

1. In this study, Bhosle and colleagues provide evidence suggesting that the secreted protein Slit2 promotes macropinocytosis in macrophages via Robo1 and RhoA. Almost all the work is conducted *in vitro* using primarily the RAW264.7 mouse cell line. Overall the data are not very original as the influence of Slits on immune cells has been extensively studied. It is already known that Slit activates RhoA and myosin in monocytes and other cell types. Most of the figures are also simple “old-fashioned” histograms and there is basically no *in vivo* data supporting the model. There are many mouse models available (Slit and Robo knockouts) that could have been used to support the model.

We thank the reviewer for their suggestions. As recommended, we have now replaced all histograms with scatter dot and box plots. As suggested, we have now also performed a series of *in vivo* studies to confirm the significance of our *in vitro* findings. (Please see Figures 5 and 6 in the revised manuscript). We did not use the SLIT2- or ROBO1- knockout mice as these mice die in the pre- or perinatal period (Plump et al., 2002; Xian et al., 2001). We have now clarified this point (Please see page- 18 in the revised manuscript).

2. The authors only used Slit2 and they have not tested the other Slits (Slit1 and Slit3) which also bind to Robo1. What supports the specificity of Slit2 activity in this model? They do not provide any *in vivo* data on the expression of these molecules in inflammatory models.

We did not examine the levels of SLIT1 since it is predominantly expressed in the central nervous system (Wu et al., 2001). Please see SLIT1 tissue expression in The Human Protein Atlas (Uhlen et al., 2015): <https://www.proteinatlas.org/ENSG00000187122-SLIT1/tissue>

Slit3 mRNA is enriched in some peripheral tissues, such as bone marrow and adipose tissue (Kim et al., 2018). *Slit2* mRNA is ubiquitously expressed in the non-neuronal tissues (Pinho et al., 2018; Wu et al., 2001; Zhao et al., 2014). We used a recently validated ELISA kit to measure SLIT2 protein levels in mouse sera and in the peritoneal membrane samples (please see Fig. 5G in the revised manuscript) (Yao et al., 2019; Zeng et al., 2018). We have now clarified these points (Please see pages- 12 and 18 in the revised manuscript). When appropriately sensitive reagents become available, future studies could investigate the effects of endogenous SLIT3 (and possibly SLIT1) protein in modulating macrophage macropinocytosis.

3. Importantly, a previous study (not cited) showed that Slit3 influences monocyte migration and the effect on cell spreading appear similar.

Geutskens et al., J Immunol. 2010 Dec 15; 185(12):7691-8.

The chemorepellent Slit3 promotes monocyte migration.

We thank the reviewer for raising this point. As the reviewer notes, the study by Geutskens *et al* demonstrated that SLIT3 increases monocyte rounding *in vitro* (Geutskens et al., 2010). However, the underlying molecular mechanism is not known. In contrast to SLIT3 which acts a chemoattractant for immune cells, SLIT2 acts as a chemorepellent for immune cells, including for monocytes/macrophages *in vitro* (Kanellis et al., 2004) and *in vivo* (Mukovozov et al., 2015). In line with the previous studies, in our current study, we found that NSlit2 inhibited MDP-induced cell migration *in vivo* (Supplementary Fig. 5A). By simultaneously inhibiting cell migration and

macropinocytosis, SLIT2 challenges the conventional notion that signals which promote cell migration inhibit macropinocytosis and vice versa. We have now cited the paper by Geutskens *et al*, and have discussed these important points in the revised version of the manuscript (Please see pages- 15-16 of the revised manuscript).

4. Likewise, why do they assume that Robo1 is the only receptor involved? Does the cell lines used (RAW264.7 and HEK293T) and human macrophage and BMDMs only express this Slit receptor?

We appreciate the reviewer raising this important point. Of the 4 Roundabout receptors (ROBO1-4), mammalian ROBO3 has poor binding affinity for Slit proteins (Zelina *et al.*, 2014) and ROBO4 is exclusively expressed in endothelial cells (Huminiacki *et al.*, 2002; Tanaka *et al.*, 2018). We have also modified the manuscript to clarify these important points (see page- 4 of the revised manuscript). Moreover, Kim BJ *et al.* have recently reported that primary murine BMDM only express ROBO1 and ROBO3 receptors (Kim *et al.*, 2018; Park *et al.*, 2019). In line with their findings, we have now confirmed that *Robo1* mRNA, but not *Robo2* mRNA, is expressed in RAW264.7 cells and murine BMDM. These new data are now presented in Fig. 1A of the revised manuscript. We also examined and confirmed that ROBO1 protein is present in RAW264.7 and primary human macrophages (Supplementary Fig. 1A).

5. The authors failed/neglected to cite many papers (sometimes in other cellular models) that already provided similar data (and often used the same drugs or blocking agents, such as blebbistatin or Y-27632), in particular on RhoA. This is not acceptable.

We are sorry that the reviewer felt that we neglected to cite published papers that presented data similar to ours. This was certainly not our intention. As we point out in the Discussion section, SLIT2-ROBO1 signaling has different effects on the activity of Rho-family GTPases based on the cell type and tissue microenvironments, and for the sake of brevity, we attempted to cite only work that we believed to be directly comparable. Below, for each of the papers mentioned, we have attempted to clarify our initial reasoning in this regard. However, we agree that a more fulsome discussion of each of these papers is warranted, and we have now done so in the revised version of the manuscript. “Differential effects of Slit2 to activate or inactivate RhoA in different cells may depend on the relative expression of SLIT2-ROBO1 effector GAPs, such as srGAPs and MYO9B” (please see page- 15 of the manuscript).

1. Wang *et al.*, *Glia*. 2013 May;61(5):710-23. Repulsive migration of Schwann cells induced by Slit-2 through Ca²⁺-dependent RhoA-myosin signaling.

In this work, Wang *et al.* demonstrated that in presence of the ROCK1/2 inhibitor, Y-27632, SLIT2 still induced the collapse of the leading front of Schwann cells. However, similar treatment with Y-27632 prevented the reversal of translocation of soma. However, any direct association between a SLIT2-induced increase in intracellular calcium and Rho activity was not investigated (please see page- 15 of the revised manuscript). In our study, we used multiple approaches to confirm our findings. In addition to using pharmacological inhibitors of Rho signaling, which can have off-target effects, we also used knockdown of RhoA in macrophages to validate our findings. We have enhanced the discussion to include these important points.

2. Liu et al. J Biol Chem. 2012 May 18;287(21):17503-16. Slit2 regulates the dispersal of oligodendrocyte precursor cells via Fyn/RhoA signaling.

Fyn kinase is a member of Src family of conserved tyrosine kinases. Liu *et al.* demonstrated that Fyn inactivation occurs following prolonged exposure (60 min) to Slit2 in oligodendrocyte precursor cells but not in mature oligodendrocytes, even though Fyn kinase is ubiquitously expressed in both cell types. In our study, we observed a robust increase in RhoA activation after short-term exposure (15 min) to NSlit2 (Fig 2B). However, we observed that the same exposure to Slit2 failed to produce any changes in activity of Fyn or total Src (Supplementary Fig. 3C) in macrophages. We have now presented these data in Supplementary Fig. 3B-C, respectively, of the revised manuscript. Furthermore, we have expanded the discussion section to clarify these points (please see pages- 8 and 15 of the revised manuscript).

3. Backer et al., Development. 2018 Oct 2;145(19). Trio GEF mediates RhoA activation downstream of Slit2 and coordinates telencephalic wiring.

In line with the reviewer's comment, we investigated the role of TRIO GEF in mediating activation of RhoA downstream of SLIT2. We examined expression of *Trio* mRNA in both primary BMDM and cultured RAW264.7 cells. We did not detect any *Trio* mRNA expression in primary BMDM, but detected very low levels of expression in RAW264.7 cells (Supplementary Fig. 4A). Our results are in keeping with another report by van Rijssel, J. *et al.* (van Rijssel and van Buul, 2012). Accordingly, we reasoned that while TRIO activation might partially contribute to the observed phenotype in RAW264.7 cells, it cannot explain the NSlit2-induced cytoskeletal changes we observed in primary human and murine macrophages (please see Supplementary Fig. 1C-D and Fig. 2A). These new data are now presented in Supplementary Fig. 3A of the revised manuscript, and we have now included discussion of the potential role of TRIO on pages- 8 and 15 of the revised manuscript.

Geutskens et al., J Immunol. 2010 Dec 15;185(12):7691-8.

The chemorepellent Slit3 promotes monocyte migration.

In the study by Geutskens *et al.* the Slit family member, Slit3, was shown to induce activation of RhoA, but no specific molecular mechanism was identified. In contrast, in our study, we examined the links between Slit2 and RhoA, and showed that myosin Ixb (MYO9B), a RhoA specific GAP, which is highly expressed in immune cells, acts as a direct downstream effector of SLIT2-ROBO1 signaling in macrophages. Additionally, by exogenously adding MYO9B-GAP in HEK293T cells which do not express the protein, we show that these cells round up upon exposure to NSlit2 (Supplementary Fig. 3F). We have now expanded the Discussion section to clarify the important differences between the two studies (please see page- 15 of the revised manuscript).

4. Guan et al., Cell. 2007 Apr 20;129(2):385-95. Long-range Ca²⁺ signaling from growth cone to soma mediates reversal of neuronal migration induced by slit-2.

Guan *et al.* report that SLIT2-induced cellular calcium changes regulate the spatial distribution (front end vs. back end of the cell) of active RhoA to control the neuronal migration. In their assay, total RhoA activity was downregulated by SLIT2 treatment (please see Fig. 4F in the cited reference) (Guan *et al.*, 2007). This is opposite to what we observe in macrophages when treated with NSlit2. We hypothesize that this might be due to immune cell-specific expression of the RhoA GAP,

MYO9B. We have now expanded the discussion to include this reference (please see page- 15 of the revised manuscript).

As mentioned above, our intention was not to neglect citing papers that present data similar to ours. Rather, for the sake of brevity and clarity, we initially chose to focus on work we felt was directly comparable to ours. We do agree with the reviewer that more fulsome consideration of the similarities and differences between the highlighted papers and the work of our group leads to a much richer, in depth discussion and understanding. Accordingly, and in accordance with the reviewer's suggestions, we have now amended the manuscript to discuss all of the points above (please see our detailed response to Comment #5).

6. What is the “shape factor”? This is rather unclear from the results and methods and they should not expect the readers to look at the Volocity manual to find out what it is.

As recommended and to enhance clarity, we have now included a detailed explanation regarding the “shape factor” (please see Methods of the revised version of the manuscript).

7. As said before many related papers are completely ignored

Zhao et al., J Immunol. 2014 Jan 1;192(1):385-93. Slit2-Robo4 pathway modulates lipopolysaccharide-induced endothelial inflammation and its expression is dysregulated during endotoxemia.

This paper show that Slit2 represses inflammatory responses by inhibiting the Pyk2-NF-κB pathway downstream of LPS-TLR4.

We thank the reviewer for raising this point. As mentioned in our response to Comment #4, ROBO4 is an endothelial cell-specific Roundabout receptor. In our study, we focused on modulation of the NOD2 signaling by SLIT2-ROBO1 in macrophages. NOD2 is exclusively found intracellularly and not on the cell surface. MDP, a NOD2-specific agonist, has been previously shown to enter primary human macrophages via constitutive macropinocytosis to exert its biological effects (Canton et al., 2016). In our study, we show that exposure to Slit2 reduces uptake of MDP by macrophages *in vitro* and *in vivo*, and consequently, that Slit2 attenuates MDP-induced NOD2 activation.

- Exposure to NSlit2 reduces MDP-FITC uptake by primary BMDMs (Fig. 5A-B)
- NSlit2 treatment attenuates MDP-induced NF-κB activation *in vitro* (Fig. 5C-D).
- New result- Using periodate-induced sterile peritonitis murine model, we show that NSlit2 reduced MDP-induced increase serum CXCL1 as effectively as EIPA (macropinocytosis inhibitor) treatment (Fig. 5E-F).
- We blocked endogenous SLIT2 action with treatment with Robo1N which augmented MDP-induced increase in serum CXCL1 and this effect was reversed by EIPA (Fig. 5H). Our results suggest that NSlit2 acts by limiting MDP entry into the cells via macropinocytosis.

We have now included in the discussion consideration of the points raised by the reviewer (please see pages- 17-18 of the revised manuscript).

Reviewer #3 (Remarks to the Author):**Bhosle et al**

I felt this manuscript delivered significantly less than was promised in the title and abstract. Macropinocytosis has emerged as an important endocytic pathway in various cell types and while activators have been described (growth factors, TLR ligands....) the authors are broadly correct in stating that no endogenous/physiological inhibitors are known. The authors claim that the Slit2 ligand upstream of Robo1 signalling is such a physiological inhibitor in macrophages. What is actually shown is that injection of a recombinant fragment of Slit2 (NSlit2) reduces constitutive endocytosis by peritoneal macrophages. Macropinocytosis is inferred by the large size of the endosomes observed. However, the claim that Slit2/Robo1 signalling is a physiological inhibitor of macropinocytosis is not exemplified by a physiological scenario where macropinocytosis *in vivo* is regulated by endogenous Slit2 as opposed to exogenous Slit2.

We thank the reviewer for this very helpful feedback. In the manuscript we originally submitted, we investigated the effect of exogenously administered NSlit2 on macropinocytosis *in vivo* by studying the uptake of 70kDa dextran by peritoneal macrophages. We have now greatly expanded the scope of this work to assess the regulation of macropinocytosis by endogenous NSlit2 *in vivo*, to measure the endogenous levels of SLIT2 protein in the peritoneum, and to block the effect of endogenous SLIT2 *in vivo*. As mentioned in our response to Comment #1 from Reviewer 2, SLIT2 knockout mice are not viable (Plump et al., 2002). Furthermore, due to a lack of commercially-available validated antibodies, levels of SLIT2 protein in the extra-neuronal tissues have not extensively been explored. Using a newly available, recently validated ELISA assay (Yao et al., 2019; Zeng et al., 2018), we measured the levels of SLIT2 protein in murine tissue samples, namely, serum and peritoneal membrane. We also performed new experiments in which we showed that MDP administered intraperitoneally underwent macropinocytosis by peritoneal macrophages, causing them to produce the chemokine, CXCL1, which is then released into the systemic circulation (Magalhaes et al., 2008). When we administered Robo1N, a soluble N-terminal fragment of the ROBO1 receptor to block the endogenous SLIT2 present, we observed an augmented response to MDP, with significantly higher levels of serum CXCL1. Using the ELISA, we further measured levels of SLIT2 in the peritoneal membrane, and detected levels ten-fold higher than levels detected in the serum. These new data are now presented in Fig. 5G and page- 18 of the revised manuscript. Overall, these results suggest that endogenous SLIT2 present locally (in this instance, in the peritoneal membrane), serves to limit macropinocytosis by macrophages *in vivo*, thereby limiting their inflammatory activation and consequent production of the inflammatory chemokine, CXCL1. We have also expanded the Discussion to consider the biologic implications of NSlit2-induced modulation of macropinocytosis (please see pages- 17-18 of the revised manuscript).

Main points

1. In spite of the focus on macropinocytosis in the title and abstract, most of the paper deals with a macrophage spreading/rounding phenotype induced by the Slit2 fragment. This aspect is hardly referred to in the Abstract. Macropinocytosis per se is only studied in 1 of the 5 main Figures (Figure 4), — 2 if you include Figure 5 which relates to NOD signalling following putative macropinocytic uptake of the MDP ligand. The Abstract does not accurately reflect the data presented in the study.

We agree with the reviewer's comment. We have now performed many new experiments that examine in much more depth the role of NSlit2 in inhibiting macropinocytosis. We showed that NSlit2 can prevent growth and reduce viability of ROBO1-expressing KRAS-transformed cancer cells that rely on macropinocytosis for nutrient uptake (Commisso et al., 2013; Lee et al., 2019). As described in our response to Comment #1 above, we also discovered that endogenous SLIT2 present locally (in this instance, in the peritoneal membrane), serves to limit macropinocytosis by macrophages *in vivo*, thereby limiting their inflammatory activation and consequent production of the inflammatory chemokine, CXCL1. These new data are presented in Fig. 5, 6, and Supplementary Fig. 5, and on pages- 12-13 of the revised manuscript. We have also rewritten the abstract to incorporate these new findings.

2. NSlit2 triggers a cell rounding phenotype (Figure 1) via RhoA activation (Figure 2). It is good to see some mechanistic basis for these effects of active Rho on cell spreading – the authors find evidence for RhoA driven formin activation (though the inhibitor used, SMIFH2 has also been reported to have other effects e.g. on p53) but how is RhoA activated by NSlit2?

We thank the reviewer for raising this important point. As the reviewer points out, SMIFH2 has been reported to have off-target effects on p53 which also affect the cellular cytoskeleton. However, Isogai T. *et al.* demonstrated that the effects on p53 are minimum at SMIFH2 concentrations below 25 μ M and treatment durations less than 1 h (Isogai et al., 2015). For this specific reason, we used SMIFH2 at a concentration 10 μ M for a duration of 30 minutes (please see Methods in the revised manuscript). In addition, we used specific siRNA targeting RhoA to confirm our findings by modulating activity and expression of RhoA. MYO9B has been recently implicated in the regulation of constitutive activity of RhoA in osteosarcoma cells to generate spontaneous cell contractility (Graessl et al., 2017). Future study will investigate if similar function of MYO9B exists in macrophages.

3. The authors state in the Discussion that inhibition of Myo9b (contains a RhoGAP) is the mechanism of activation of RhoA. Really? What is shown (in Figure 3) is vector-driven expression of Myo9b GAP and its effects on Nslit2/Robo signalling. I could see no analysis of endogenous Myo9b and its GAP activity. Surely this is essential to make any claim about how RhoA is activated in this setting?

We thank the reviewer for this important note. The endogenous effects of MYO9B in monocytes/macrophages have been previously investigated by Hanley PJ *et al.* who demonstrated that MYO9B-deficient monocytes/macrophages are round in morphology and migrate less towards chemoattractants due to failure to generate lamellipodia (Hanley et al., 2010). We and others have previously reported the chemorepellent action of Slit2 on monocytes and macrophages *in vitro* as well as *in vivo* (Kanellis *et al.*, 2004; Mukovozov *et al.*, 2015). Therefore, NSlit2's actions on macrophages phenocopy the MYO9B deficiency. Moreover, structural analyses reported by Kong *et al.* previously demonstrated that the RhoGAP domain of MYO9B specifically recognizes RhoA and inhibits its activity (Kong et al., 2015). In our current study, we show that overexpression of the MYO9B-GAP domain overcomes and reverses the effects of NSlit2 on cytoskeletal rearrangement in macrophages (please see Fig. 3A-C). In contrast, we found that HEK293T cells have low levels of MYO9B protein as compared to macrophages (Supplementary Fig. 3D), and consequently, HEK293T cells round up when exposed to Slit2 only after the exogenous expression of MYO9B-

GAP (Supplementary Fig. 3G). Taken together, these results suggest that Slit2-induced cytoskeletal changes are predominantly due to inactivation of MYO9B-GAP which in turn activates RhoA. We have enhanced the Discussion section to clarify our results and to discuss the supporting literature in more detail (please see page- 16 of the revised manuscript).

4. This is important because a mechanistic basis for Slit2-mediated activation of RhoA was already addressed in a paper last year by Backer et al in Development (<http://dev.biologists.org/content/145/19/dev153692.long>). Those authors showed that Trio was the GEF responsible for RhoA activation downstream of Slit/Robo signalling. Could this be relevant in macrophages? The authors do not address this or refer to the Backer study. Generally I found the section describing the experiments with Myo9b somewhat confusing and I may have wrongly got the impression that RhoGAP Myo9B has opposite effects in RAW and HEK293 cells: NSlit2-induced rounding was inhibited in RAW cells (as expected from the previous RhoA activation data) by Myo9b but in Robo1 transfected HEK293 expressing Myo9B GAP domain rounding was increased? Is that right? I apologise if I have missed something but this section generally was unclear and did not seem to support the notion that Slit/Robo signalling regulates endogenous Myo9b GAP activity.

We apologize for the confusion regarding effects of SLIT2 on macrophages vs HEK293T cells. Please see our response to Comment #3 above. In addition, we have rewritten the discussion to clarify our findings and interpretation (please see pages- 9 and 16 of the revised manuscript). As suggested by both Reviewers 2 and 3, we have now investigated the role of TRIO GEF in mediating activation of RhoA downstream of Slit2. We did not detect *Trio* mRNA expression in BMDM but detected very low levels of expression in RAW264.7 cells (please see Supplementary Fig. 4A). Our results are in line with another report (van Rijssel and van Buul, 2012). Accordingly, we reasoned that while TRIO activation might contribute to the observed phenotype in RAW264.7 cells, it cannot explain the Slit2-induced cytoskeletal changes we observed in primary human (Supplementary Fig. 1C-D) and murine (Fig. 2A) macrophages. We have now included discussion of the potential role of TRIO GEF on page- 15 of the revised manuscript.

5. The authors appear to rule out Rho-driven effects on Rac activation state as a mechanism to explain the cell rounding phenotype but do not investigate whether suppression of Rac1/2 activation might nonetheless explain the NSlit2 effect on macropinocytosis. There is a well described reciprocal antagonism of Rac and Rho activity in cells so it is quite possible (notwithstanding the result obtained in the cell rounding experiments) that RhoA activation induced by the Slit ligand shuts down Rac activity and this is the reason for the suppression of macropinocytosis. This was not investigated.

We thank the reviewer for raising this important point, and agree that several studies have reported SLIT2-induced Rac1/2-inactivation in different cell types, including in RAW 264.7 cells (Kanellis et al., 2004). As suggested, we have now investigated the differential roles of Rac inhibition and RhoA activation in modulating effects induced by NSlit2. We examined effects of NSlit2 to inhibit Rac in the presence and absence of the Rho inhibitor, TAT-C3, which does not inhibit Rac or Cdc42. We found that Rho inhibition alone activated Rac, but NSlit2 attenuated this Rac activation which suggests that it occurs, at least partly, independent of the MYO9B-mediated effect on Rho activity. These new data are presented in Supplementary Fig. 4C and on pages- 10 and 17 of the revised manuscript.

6. Are we looking exclusively at macropinocytosis in the *in vitro* and *in vivo* experiments described? We are reliant on the size of the endosomes shown to conclude that these are macropinosomes. The authors do not show the background dextran uptake in the absence of Csf1, the inducer of macropinocytosis used. A helpful experiment would be to show Csf1 stimulation of macropinocytosis and suppression by a well-established inhibitor (e.g. amiloride) alongside Nslit2 in the *in vitro* experiments.

We thank the reviewer for this helpful suggestion. As recommended, we used EIPA, a well-established amiloride derivative, to inhibit macropinocytosis *in vitro* and *in vivo* (please see Figures 5E, 5H, 6A, and 6D). Furthermore, Canton *et al.* demonstrated that Csf1 treatment selectively increases the number of macropinosomes larger than 3 μm in diameter in macrophages (Canton *et al.*, 2016). Accordingly, in our experiments, we counted all pinosomes larger than 0.2 μm in diameter (please see 'Methods' in the revised manuscript), which would include CSF1-induced macropinosomes.

Other points:

1. The text is sometimes hard to follow with the flow being interrupted with bracketed text containing P values and 'this condition versus that condition' explanations. The paper would be easier to read if all this had been in the Figure legends and not the text.

None of the pages or Figures are numbered.

We apologize for the oversight. We have numbered all pages and lines in the revised manuscript. We have also included the relevant comparisons and p values in the Figure Legends section.

2. There are no independent size markers in Fig 1A. Is the difference in human v mouse Robo1 migration due to different sized proteins?

Human ROBO1 protein has 3 confirmed isoforms (a, b, d) of 1651, 1606, and 1551 amino acids (aa), respectively, while the murine orthologue only has one confirmed isoform of 1593 aa. As suggested, we have now added the size markers in Supplementary Fig. 1A.

3. Why was full length Slit2 never tested?

All recombinant Slit2 proteins (NSlit2, CSlit2 and Slit2 Δ D2) were prepared in our laboratory and verified to be endotoxin-free prior to use in *in vitro* and *in vivo* experiments (please see Supplementary Table- 1). Full-length SLIT2 protein is known to be cleaved into N-terminal and C-terminal fragments *in vivo* (Nguyen Ba-Charvet *et al.*, 2001). Full length SLIT2 is approximately 200 kDa, whereas NSlit2 is 140 kDa, CSlit2 is 50 kDa (Chedotal, 2007), and are easier to purify. In both *in vitro* and *in vivo* studies in immune cells, we previously demonstrated that full-length SLIT2 and NSlit2 produce similar effects (Chaturvedi *et al.*, 2013).

4. There are no controls for siRNA knockdown of Rac and Rho. To exclude off-target effects it is generally considered essential to show 2 different siRNAs and to 'rescue' the phenotype by transfection of an siRNA resistant construct for the target gene. Neither of these controls are shown.

In this study, we did not use siRNA against Rac proteins. Instead, we used primary BMDM obtained from Rac1/2 double knockout mice, which have been validated in previous studies (Sun *et al.*, 2004; Wang *et al.*, 2008) (see Fig. 2A). To verify these results, we used CK-666, a pharmacologic inhibitor of the Arp2/3 complex which is a downstream effector of Rac (Nolen *et al.*, 2009) (please see Supplementary Fig. 2B in the revised manuscript).

To verify results from experiments using siRNA knockdown of RhoA, we also used an independent approach, namely pharmacological inhibition of RhoA using TAT-C3 (please see Fig. 2C-D).

5. Expression of active Rac is shown not to rescue the rounding phenotype induced by Nslit2 (Figure 2A). No data are presented showing how much Q61L Rac was expressed (relative to endogenous Rac) and in what proportion of cells. Also this expt is only meaningful if, in control cells (i.e non NSlit2 treated macrophages), Q67L Rac1 is shown to drive a response.

Following transfection of RAW264.7 cells with a plasmid encoding GFP-tagged constitutively active Rac1 (Q61L), the corresponding protein was produced by approximately one-third of cells. We used GFP only transfected cells as a control in this experiment (see new Supplementary Fig. 2A and page- 6 of the revised manuscript).

6. There are large differences in cell surface area between some expts? E.g. Fig 1C 1200 μ 2 to 600 reduction on Nslit2 exposure whereas in the expt where the Arp2/3 inhibitor is tested (Supplementary 2B) the numbers are about 350 to 120? Are the cells really 4 times larger/smaller between expts?

It is known that Rac activation is necessary for macrophage spreading, while Rho activity is needed for retraction (Ory *et al.*, 2000). Cell spreading changes induced by Arp2/3 inhibition were quantified again and they were comparable (see Supplementary Fig. 2B) to cell-spreading observed in in Rac1/2 double knockout BMDMs (please see Fig. 2A in the revised manuscript).

7. Overall, for the reasons stated above I felt that there was not a strong and clear enough message for a high ranked journal. The claim that Slit2/Robo1 signalling is a physiological inhibitor of macropinocytosis is not supported by a physiological scenario where this signalling pathway operates. For example, constitutive macropinocytosis in dendritic cells is progressively shut down upon physiological maturation as shown by many labs. Might that be mediated by Slit2/Robo signalling? Since macropinocytosis is often an acutely stimulated process it is debateable whether physiological inhibitors are actually needed: attenuation may be simply due to the removal of the stimulus or feedback inhibition of the activation pathway. The effects of Slit2/Robo signalling on macrophages described here will nonetheless be of interest to a more specialised audience, for example in a dedicated cell biology journal.

In the initial version of the manuscript, we did not examine the physiologic or pathophysiologic importance of SLIT2 in inhibition of macrophage macropinocytosis. We have now performed many new experiments to explore these concepts further. We showed that NSlit2 can prevent growth and reduce viability of ROBO1-expressing cancer cells that rely on macropinocytosis for nutrient uptake (please see new Fig. 6). For these experiments, we used KRAS-transformed cancer cell-lines. Pancreatic ductal adenocarcinoma (PDAC) has a 5-year survival rate of 8% (Siegel *et al.*, 2017). The KRAS transformation, a known driver of macropinocytosis in cancer cells, is found in > 90%

PDACs (Haigis, 2017). Lee and colleagues recently demonstrated that the central tumor cores in PDAC are more nutrient-deficient than the peripheral regions. Specifically, glutamine-deprivation induces macropinocytosis in PDAC cells to promote their survival (Commisso et al., 2013; Lee et al., 2019). SLIT2 has been previously reported to be silenced in PDAC cell-lines, and conversely, high SLIT2 levels are associated with better patient survival (Gohrig et al., 2014). We here report that NSlit2 blocks macropinocytosis in KRAS-transformed PDAC cells in the glutamine-deficient environment and thereby decreases tumor-cell survival. Therefore, SLIT2, which inhibits neural invasion, metastasis, and macropinocytosis in PDAC, could be a promising anti-cancer therapeutic target (Göhrig et al., 2014). These data are presented in Fig. 6 and discussed on page- 19 of the revised manuscript.

New experiments we performed revealed that endogenous SLIT2 present locally (in this instance, in the mouse peritoneal membrane), serves to limit macropinocytosis by peritoneal macrophages *in vivo*, thereby limiting their inflammatory activation and consequent production of the inflammatory chemokine, CXCL1. These new data are presented in Fig. 5E-H, and on pages- 12 of the revised manuscript. We have also rewritten the abstract to incorporate these new findings. Overall, SLIT2 challenges the conventional notion that signals that inhibit cell migration do not inhibit macropinocytosis, and vice versa. Our studies also define a novel role for SLIT2 in inhibiting macropinocytosis that stimulates cancer cell growth, and that leads to inflammatory chemokine production by macrophages. We have now rewritten the 'Discussion' section of the manuscript to emphasize these important points (please see pages- 18-19 of the revised manuscript).

References:

- Canton, J., Schlam, D., Breuer, C., Gutschow, M., Glogauer, M., and Grinstein, S. (2016). Calcium-sensing receptors signal constitutive macropinocytosis and facilitate the uptake of NOD2 ligands in macrophages. *Nat Commun* 7, 11284.
- Chaturvedi, S., Yuen, D.A., Bajwa, A., Huang, Y.W., Sokollik, C., Huang, L., Lam, G.Y., Tole, S., Liu, G.Y., Pan, J., *et al.* (2013). Slit2 prevents neutrophil recruitment and renal ischemia-reperfusion injury. *J Am Soc Nephrol* 24, 1274-1287.
- Chedotal, A. (2007). Slits and their receptors. *Adv Exp Med Biol* 621, 65-80.
- Commisso, C., Davidson, S.M., Soydaner-Azeloglu, R.G., Parker, S.J., Kamphorst, J.J., Hackett, S., Grabocka, E., Nofal, M., Drebin, J.A., Thompson, C.B., *et al.* (2013). Macropinocytosis of protein is an amino acid supply route in Ras-transformed cells. *Nature* 497, 633-637.
- Geutskens, S.B., Hordijk, P.L., and van Hennik, P.B. (2010). The chemorepellent Slit3 promotes monocyte migration. *J Immunol* 185, 7691-7698.
- Gohrig, A., Detjen, K.M., Hilfenhaus, G., Korner, J.L., Welzel, M., Arsenic, R., Schmuck, R., Bahra, M., Wu, J.Y., Wiedenmann, B., *et al.* (2014). Axon guidance factor SLIT2 inhibits neural invasion and metastasis in pancreatic cancer. *Cancer Res* 74, 1529-1540.
- Graessl, M., Koch, J., Calderon, A., Kamps, D., Banerjee, S., Mazel, T., Schulze, N., Jungkurth, J.K., Patwardhan, R., Solouk, D., *et al.* (2017). An excitable Rho GTPase signaling network generates dynamic subcellular contraction patterns. *J Cell Biol* 216, 4271-4285.
- Guan, C.B., Xu, H.T., Jin, M., Yuan, X.B., and Poo, M.M. (2007). Long-range Ca²⁺ signaling from growth cone to soma mediates reversal of neuronal migration induced by Slit-2. *Cell* 129, 385-395.

Haigis, K.M. (2017). KRAS Alleles: The Devil Is in the Detail. *Trends Cancer* 3, 686-697.

Hanley, P.J., Xu, Y., Kronlage, M., Grobe, K., Schon, P., Song, J., Sorokin, L., Schwab, A., and Bahler, M. (2010). Motorized RhoGAP myosin IXb (Myo9b) controls cell shape and motility. *Proc Natl Acad Sci U S A* 107, 12145-12150.

Huminiecki, L., Gorn, M., Suchting, S., Poulsom, R., and Bicknell, R. (2002). Magic roundabout is a new member of the roundabout receptor family that is endothelial specific and expressed at sites of active angiogenesis. *Genomics* 79, 547-552.

Isogai, T., van der Kammen, R., and Innocenti, M. (2015). SMIFH2 has effects on Formins and p53 that perturb the cell cytoskeleton. *Sci Rep* 5, 9802.

Kanellis, J., Garcia, G.E., Li, P., Parra, G., Wilson, C.B., Rao, Y., Han, S., Smith, C.W., Johnson, R.J., Wu, J.Y., *et al.* (2004). Modulation of inflammation by slit protein in vivo in experimental crescentic glomerulonephritis. *Am J Pathol* 165, 341-352.

Kim, B.J., Lee, Y.S., Lee, S.Y., Baek, W.Y., Choi, Y.J., Moon, S.A., Lee, S.H., Kim, J.E., Chang, E.J., Kim, E.Y., *et al.* (2018). Osteoclast-secreted SLIT3 coordinates bone resorption and formation. *J Clin Invest* 128, 1429-1441.

Kong, R., Yi, F., Wen, P., Liu, J., Chen, X., Ren, J., Li, X., Shang, Y., Nie, Y., Wu, K., *et al.* (2015). Myo9b is a key player in SLIT/ROBO-mediated lung tumor suppression. *J Clin Invest* 125, 4407-4420.

Lee, S.W., Zhang, Y., Jung, M., Cruz, N., Alas, B., and Commisso, C. (2019). EGFR-Pak Signaling Selectively Regulates Glutamine Deprivation-Induced Macropinocytosis. *Dev Cell*.

Magalhaes, J.G., Fritz, J.H., Le Bourhis, L., Sellge, G., Travassos, L.H., Selvanantham, T., Girardin, S.E., Gommerman, J.L., and Philpott, D.J. (2008). Nod2-dependent Th2 polarization of antigen-specific immunity. *J Immunol* 181, 7925-7935.

Mason, F.M., Heimsath, E.G., Higgs, H.N., and Soderling, S.H. (2011). Bi-modal regulation of a formin by srGAP2. *J Biol Chem* 286, 6577-6586.

Mukovozov, I., Huang, Y.W., Zhang, Q., Liu, G.Y., Siu, A., Sokolskyy, Y., Patel, S., Hyduk, S.J., Kutryk, M.J., Cybulsky, M.I., *et al.* (2015). The Neurorepellent Slit2 Inhibits Postadhesion Stabilization of Monocytes Tethered to Vascular Endothelial Cells. *J Immunol* 195, 3334-3344.

Nguyen Ba-Charvet, K.T., Brose, K., Ma, L., Wang, K.H., Marillat, V., Sotelo, C., Tessier-Lavigne, M., and Chedotal, A. (2001). Diversity and specificity of actions of Slit2 proteolytic fragments in axon guidance. *J Neurosci* 21, 4281-4289.

Nolen, B.J., Tomasevic, N., Russell, A., Pierce, D.W., Jia, Z., McCormick, C.D., Hartman, J., Sakowicz, R., and Pollard, T.D. (2009). Characterization of two classes of small molecule inhibitors of Arp2/3 complex. *Nature* 460, 1031-1034.

Ory, S., Munari-Silem, Y., Fort, P., and Jurdic, P. (2000). Rho and Rac exert antagonistic functions on spreading of macrophage-derived multinucleated cells and are not required for actin fiber formation. *J Cell Sci* 113 (Pt 7), 1177-1188.

- Park, S.J., Lee, J.Y., Lee, S.H., Koh, J.M., and Kim, B.J. (2019). SLIT2 inhibits osteoclastogenesis and bone resorption by suppression of Cdc42 activity. *Biochem Biophys Res Commun* 514, 868-874.
- Patel, S., Huang, Y.W., Reheman, A., Pluthero, F.G., Chaturvedi, S., Mukovozov, I.M., Tole, S., Liu, G.Y., Li, L., Durocher, Y., *et al.* (2012). The cell motility modulator Slit2 is a potent inhibitor of platelet function. *Circulation* 126, 1385-1395.
- Pinho, A.V., Van Bulck, M., Chantrill, L., Arshi, M., Sklyarova, T., Herrmann, D., Vennin, C., Gallego-Ortega, D., Mawson, A., Giry-Laterriere, M., *et al.* (2018). ROBO2 is a stroma suppressor gene in the pancreas and acts via TGF-beta signalling. *Nat Commun* 9, 5083.
- Plump, A.S., Erskine, L., Sabatier, C., Brose, K., Epstein, C.J., Goodman, C.S., Mason, C.A., and Tessier-Lavigne, M. (2002). Slit1 and Slit2 cooperate to prevent premature midline crossing of retinal axons in the mouse visual system. *Neuron* 33, 219-232.
- Siegel, R.L., Miller, K.D., and Jemal, A. (2017). Cancer Statistics, 2017. *CA Cancer J Clin* 67, 7-30.
- Sun, C.X., Downey, G.P., Zhu, F., Koh, A.L., Thang, H., and Glogauer, M. (2004). Rac1 is the small GTPase responsible for regulating the neutrophil chemotaxis compass. *Blood* 104, 3758-3765.
- Sutterwala, F.S., Haasken, S., and Cassel, S.L. (2014). Mechanism of NLRP3 inflammasome activation. *Ann N Y Acad Sci* 1319, 82-95.
- Tanaka, T., Izawa, K., Maniwa, Y., Okamura, M., Okada, A., Yamaguchi, T., Shirakura, K., Maekawa, N., Matsui, H., Ishimoto, K., *et al.* (2018). ETV2-TET1/TET2 Complexes Induce Endothelial Cell-Specific Robo4 Expression via Promoter Demethylation. *Sci Rep-Uk* 8.
- Uhlen, M., Fagerberg, L., Hallstrom, B.M., Lindskog, C., Oksvold, P., Mardinoglu, A., Sivertsson, A., Kampf, C., Sjostedt, E., Asplund, A., *et al.* (2015). Proteomics. Tissue-based map of the human proteome. *Science* 347, 1260419.
- van Helden, S.F., Anthony, E.C., Dee, R., and Hordijk, P.L. (2012). Rho GTPase expression in human myeloid cells. *PLoS One* 7, e42563.
- van Rijssel, J., and van Buul, J.D. (2012). The many faces of the guanine-nucleotide exchange factor trio. *Cell Adh Migr* 6, 482-487.
- Wang, Y., Lebowitz, D., Sun, C., Thang, H., Grynopas, M.D., and Glogauer, M. (2008). Identifying the relative contributions of Rac1 and Rac2 to osteoclastogenesis. *J Bone Miner Res* 23, 260-270.
- Werbowetski-Ogilvie, T.E., Seyed Sadr, M., Jabado, N., Angers-Loustau, A., Agar, N.Y., Wu, J., Bjerkgvig, R., Antel, J.P., Faury, D., Rao, Y., *et al.* (2006). Inhibition of medulloblastoma cell invasion by Slit. *Oncogene* 25, 5103-5112.
- Wu, J.Y., Feng, L., Park, H.T., Havlioglu, N., Wen, L., Tang, H., Bacon, K.B., Jiang, Z., Zhang, X., and Rao, Y. (2001). The neuronal repellent Slit inhibits leukocyte chemotaxis induced by chemotactic factors. *Nature* 410, 948-952.
- Xian, J., Clark, K.J., Fordham, R., Pannell, R., Rabbitts, T.H., and Rabbitts, P.H. (2001). Inadequate lung development and bronchial hyperplasia in mice with a targeted deletion in the Dutt1/Robo1 gene. *Proc Natl Acad Sci U S A* 98, 15062-15066.

Yao, Y., Zhou, Z., Li, L., Li, J., Huang, L., Li, J., Qi, C., Zheng, L., Wang, L., and Zhang, Q.Q. (2019). Activation of Slit2/Robo1 Signaling Promotes Tumor Metastasis in Colorectal Carcinoma through Activation of the TGF-beta/Smads Pathway. *Cells* 8.

Zelina, P., Blockus, H., Zagar, Y., Peres, A., Friocourt, F., Wu, Z., Rama, N., Fouquet, C., Hohenester, E., Tessier-Lavigne, M., *et al.* (2014). Signaling switch of the axon guidance receptor Robo3 during vertebrate evolution. *Neuron* 84, 1258-1272.

Zeng, Z., Wu, Y., Cao, Y., Yuan, Z., Zhang, Y., Zhang, D.Y., Hasegawa, D., Friedman, S.L., and Guo, J. (2018). Slit2-Robo2 signaling modulates the fibrogenic activity and migration of hepatic stellate cells. *Life Sci* 203, 39-47.

Zhao, H., Anand, A.R., and Ganju, R.K. (2014). Slit2-Robo4 pathway modulates lipopolysaccharide-induced endothelial inflammation and its expression is dysregulated during endotoxemia. *J Immunol* 192, 385-393.

Reviewers' comments:

Reviewer #1 (Remarks to the Author):

In this revised version of the manuscript, the authors have added a significant amount of new data. In particular the new *in vivo* data are very nice and convincingly demonstrate that the pathway they suggest is physiological. They also add data showing that the regulation of macropinocytosis by NSlit2 is conserved in some Ras-transformed cancer cells, indicating a more general role for this phenomenon.

I still find the structure of the manuscript a bit confusing however. Whilst the title, abstract and major conclusions are all about the regulation of macropinocytosis, the major mechanistic experiments are still all based on cell spreading. Whilst they have nice data to show this is via RhoA rather than Rac regulation, the relevance of this to the main message of the paper about macropinocytosis is unclear and feels a little misleading - especially as they now show that unlike spreading, the effects of NSlit2 on macropinocytosis depend as much on Rac as Rho. In my opinion, the manuscript would be significantly improved and clearer by restructuring, putting macropinocytosis first (i.e. starting with Fig.4), then after demonstrating that reciprocal regulation of RhoA and Rac regulation, using the spreading assay to dissect the RhoA regulatory mechanism (figs 1-3).

Apart from this suggestion, which I hope the authors consider seriously, they have satisfied my technical criticisms and with the increased scope and demonstration of physiological significance, I feel the paper is much improved and could be accepted for publication.

typos:

P12 l240 Supplementaty

P13 L264 macopinocytosis

L268 respectivly

L270 blokced

L273 respectivly, ROBO-expreveessing

L276 "growth of PANC-1 cells" They only show viability

Reviewer #2 (Remarks to the Author):

In their revised manuscript the authors have addressed some of my concerns and added missing references.

However I still think that the data are not solid and that the *in vivo* support is very weak.

If Slit2 acts on macropinocytosis *in vivo*, one would expect that interfering with Slit2 signaling would have some effect on myeloid cell function *in vivo*. This is not at all addressed.

I am still not convinced by the specificity of the described effect for Slit2 and Robo1. The authors can't just assume from databases that other Slits are not expressed in the body, in particular in an *in vivo* context. Many papers have shown that Slit1 and slit3 are expressed in peripheral organs. Titrating Slit2 in the culture medium with a soluble Robo1 does not prove that Slit2 acts through Robo1 in macrophages.

According to the methods, Slit2 proteins were injected at 30 nM but the authors do not say anything about the injected volume. How was the concentration (dose) determined and what is the half life of the 3 proteins injected. Slit2 should also act on vessel permeability.

The authors also need to show the CD68 staining together with the dextran.

The authors say that Robo1 KO mice are not viable, but this is not the case and several viable lines exist. See for instance *J Clin Invest.* 2018;128(4):1429-1441) where studies were performed in Robo1^{-/-} adult mice. Likewise, Slit2 conditional lines also exist and were used by many groups (see for example Pinho et al, *Nat Commun.* 2018 Nov 30;9(1):5083).

Reviewer #3 (Remarks to the Author):

This manuscript is considerably improved compared with the previous version. The data are better presented, more comprehensive and it is much easier to read. There is now at least some evidence that Slit2/Robo signalling regulates macropinocytosis and downstream effects in vivo. The experiments where Slit ligands are injected into mice are really just using the mouse as a 'test tube' and are less convincing than those where levels and effects of endogenous Slit ligand are probed. The data here are quite limited and found in Figure 5H but at least they seem to show that endogenous Slit has real effects on MDP signalling and it is plausible that this is mediated by effects on endogenous macropinocytosis of the ligand. For me this was an important new experiment.

On a minor point of clarification, the authors refer a couple of times to the stimulation of macropinocytosis by pathogen-derived ligands like LPS (e.g on page 3). This is only part of the story as anyone familiar with the dendritic cell field will tell you. The key early experiments of Lanzavecchia and colleagues using human DC showed that LPS down-regulates macropinocytosis (Sallusto et al *JExpMed*, 1995) over the long term (a finding confirmed by others) and it was only subsequently shown (by West et al *Science*, 2004) that prior to this downregulation, there is a transient stimulation of macropinocytosis by TLR ligands. This was why I previously asked whether Slit/Robo signalling might be involved in long-term suppression of macropinocytosis in dendritic cells.

Overall I felt this study was worthy of publication.

There is a typo on line 273 which suggests spell checking/proof reading is needed.

We thank all the reviewers for their valuable feedback. Based on the comments from each expert reviewer, we have again performed additional experiments and have clarified some points from the earlier version of the manuscript. Below is a point-by-point response to each of the Reviewer's comments and accompanying changes we made to the figures and text.

Reviewer #1 (Remarks to the Author):

In this revised version of the manuscript, the authors have added a significant amount of new data. In particular the new in vivo data are very nice and convincingly demonstrate that the pathway they suggest is physiological. They also add data showing that the regulation of macropinocytosis by NSlit2 is conserved in some Ras-transformed cancer cells, indicating a more general role for this phenomenon.

We thank the reviewer for these positive comments on the revised manuscript.

I still find the structure of the manuscript a bit confusing however. Whilst the title, abstract and major conclusions are all about the regulation of macropinocytosis, the major mechanistic experiments are still all based on cell spreading. Whilst they have nice data to show this is via RhoA rather than Rac regulation, the relevance of this to the main message of the paper about macropinocytosis is unclear and feels a little misleading - especially as they now show that unlike spreading, the effects of NSlit2 on macropinocytosis depend as much on Rac as Rho. In my opinion, the manuscript would be significantly improved and clearer by restructuring, putting macropinocytosis first (i.e. starting with Fig.4), then after demonstrating that reciprocal regulation of RhoA and Rac regulation, using the spreading assay to dissect the RhoA regulatory mechanism (figs 1-3).

Apart from this suggestion, which I hope the authors consider seriously, they have satisfied my technical criticisms and with the increased scope and demonstration of physiological significance, I feel the paper is much improved and could be accepted for publication.

We thank the reviewer for the suggestion. In Fig. 1-3, we explored the mechanisms underlying NSlit2-induced cytoskeletal rearrangements in macrophages. We found that these cytoskeletal changes depend on NSlit2-ROBO1-induced activation of RhoA. In Fig. 4, before exploring the effects of SLIT2 on macropinocytosis, we investigated the potential role of RhoA in the regulation of macropinocytosis, which remains poorly defined (Pertz et al., 2006; Swanson, 2008). We found that RhoA activation inhibits macropinocytosis, while the inactivation of RhoA stimulates macropinocytosis (please see Fig. 4A-C in the revised manuscript). Therefore, Fig. 4 logically follows the preceding Fig. 1-3 in the sequential examination of RhoA's effects on constitutive macropinocytosis in macrophages.

We and others have previously demonstrated that SLIT2 inhibits chemokine-induced Rac1 (and Cdc42) activation in monocytes (Mukovozov et al., 2015), in RAW264.7 macrophages (Kanellis et al., 2004), and in bone marrow-derived macrophages (BMDM) (Park et al., 2019). We previously reported that SLIT2 inhibits post-adhesion stabilization of monocytes tethered to activated endothelial cells and this depends on Rac1 inhibition (Mukovozov et al., 2015). However, underlying mechanism responsible for SLIT-mediated RhoA activation in immune cells had not been explored.

As stated in the Discussion section of the manuscript, differences in the relative contributions of NSlit2-induced Rac inactivation and Rho activation in macrophages might be attributed to endogenous levels of Rho-family GTPases, with macrophages expressing more RhoA than Rac1/2 (van Helden et al., 2012), as well as to the differences in endogenous levels of Slit2/Robo1 effector proteins (Myo9b vs Srgap2) in these cells. We have reworded this section in the Discussion to clarify these points (please see page 16 in the revised manuscript).

We also found that siRNA-mediated RhoA knockdown partially reversed NSlit2-mediated inhibition of macropinocytosis (see Supplementary Fig. 4A-B in the revised manuscript). Therefore, Fig. 4 is connected to and builds upon the preceding data presented in Fig. 1-3 in the manuscript. We also found that NSlit2-mediated inactivation of Rac did not depend on RhoA activation (Supplementary Fig. 4C). However, there are more than 20 Rho family members in mammalian genomes and most of them are expressed in macrophages (van Helden et al., 2012). Relative contributions of the different Rho-family small GTPases in the regulation of macropinocytosis have not yet been fully elucidated. Although our findings suggest that opposing effects on RhoA and Rac1 participate in NSlit2-mediated inhibition of macropinocytosis, at present, we cannot definitively rule out a minor role for other small GTPases. We have clarified this important point in the Discussion section (please see page 18 in the revised manuscript).

For all of the reasons, we feel that the order of figures presented allows newer data and concepts to build upon the preceding data presented.

Typos:

P12 I240 Supplementaty

P13 L264 macopinocytosis

L268 respectivly

L270 blokced

L273 respectivly, ROBO-expreveessing

L276 "growth of PANC-1 cells" They only show viability

We apologize for these typographical errors. We have now corrected these errors in the revised manuscript. Please see pages 12-13 of the revised manuscript.

Reviewer #2 (Remarks to the Author):

In their revised manuscript, the authors have addressed some of my concerns and added missing references.

However I still think that the data are not solid and that the in vivo support is very weak. If Slit2 acts on macropinocytosis in vivo, one would expect that interfering with Slit2 signaling would have some effect on myeloid cell function in vivo. This is not at all addressed.

We thank the reviewer for raising this important point. As the reviewer suggests, we and others have previously demonstrated that SLIT2 inhibits directed cell migration of several myeloid cell types *in vivo*. These include neutrophils (Chaturvedi et al., 2013; Tole et al., 2009; Ye et al., 2010), monocytes/macrophages (Kanellis et al., 2004; Mukovozov et al., 2015) and dendritic cells (Guan et al., 2003). We have now included these references in the Introduction section. Please see page 4 of the revised manuscript.

In line with the previous reports, in the present study, we found that NSlit2 treatment significantly reduced MDP-induced myeloid cell migration into the peritoneal cavity (Supplementary Fig. 5A). Please see page 12 of the revised manuscript.

In previous studies by our group, some of the effects of SLIT2 on myeloid cell function have been assessed. Indeed, we reported that SLIT2-treated mice clear bacterial infections with *E. coli* and *L. monocytogenes* as efficiently as the vehicle treated counterparts *in vivo* (Chaturvedi et al., 2013). We and others have also demonstrated that exposure to NSlit2 enhances *in vitro* neutrophil killing of

S. aureus, and generation of reactive oxygen species (ROS) but does not adversely affect phagocytosis in neutrophils (Chaturvedi et al., 2013; Wu et al., 2001).

A more detailed understanding of the actions of SLIT2 on *in vivo* myeloid cell functions, such as phagocytosis and ROS production, will be investigated in future studies. We have now clarified these important points in the discussion section (please see page 15 of the revised manuscript).

I am still not convinced by the specificity of the described effect for Slit2 and Robo1. The authors can't just assume from databases that other Slits are not expressed in the body, in particular in an *in vivo* context. Many papers have shown that Slit1 and slit3 are expressed in peripheral organs.

We thank the reviewer for raising this very important point. Wu *et al.* demonstrated that *Slit1* messenger RNA (mRNA) is not detected in peripheral organs (Wu et al., 2001). In recent years, several independent groups have reported the detection of *Slit2* and *Slit3* mRNA in peripheral tissues (Kim et al., 2018; Pinho et al., 2018; Wang et al., 2020). Burgstaller et al. reported the presence of SLIT2 and SLIT3 proteins in healthy murine lungs by Mass Spectrometry, but interestingly, they did not detect SLIT1 (Burgstaller et al., 2017).

In the past, accurate quantification of SLIT2 and SLIT3 protein levels in peripheral tissues has been prevented by the lack of sensitive reagents, including commercially-available antibodies. The recent availability of commercially-available ELISA kits has allowed independent groups to measure SLIT2 and SLIT3 protein levels in serum (human and murine) and bone-marrow (murine) (Chang et al., 2015; Xu et al., 2018; Yao et al., 2019; Zeng et al., 2018). To date, SLIT protein levels in peripheral tissues have not been explored.

In order to detect and quantify the levels of SLIT2 and SLIT3 in the peritoneum, we used the above-mentioned ELISA kits. We found that SLIT2 is approximately 10 times more concentrated in the peritoneum as compared to serum; while SLIT3 levels are similar in both tissues. These data have now been added to the revised manuscript. Please see Fig. 5G, and **new** Supplementary Fig. 5C and 5D of the revised manuscript. Please also see page 12 of the revised manuscript. Our results are in line with the study by Kim BJ et al. which reported that *Slit3* mRNA is enriched in specific peripheral tissues such as bone marrow, while *Slit2* is ubiquitously expressed (Kim et al., 2018).

We were unable to measure SLIT1 protein levels due to the lack of validated reagents. When such reagents become available, a new study can be undertaken. In any event, we anticipate that levels of SLIT1 protein levels would be quite low given the observation that *Slit1* mRNA is not detected in peripheral organs (Wu et al., 2001). We have now clarified this point in the text of the manuscript (please see page 19 of the revised manuscript).

Titration of Slit2 in the culture medium with a soluble Robo1 does not prove that Slit2 acts through Robo1 in macrophages.

We appreciate the reviewer's comment. Accordingly, we used several complementary approaches to dissect the role of ROBO1 in mediating Slit2's actions on inhibition of macropinocytosis in macrophages and RAS-transformed cancer cells.

First, in keeping with previous studies, we detected *Robo1* but not *Robo2* mRNA in murine macrophages (Fig. 1A) (Geutskens et al., 2010; Kim et al., 2018). We confirmed the presence of ROBO1 protein in human and murine macrophages (Supplementary Fig. 1A). Kim BJ et al. demonstrated the presence of ROBO3 in primary murine bone-marrow-derived macrophages (BMDM) (Kim et al., 2018). However, ROBO3 has been shown to not bind Slit proteins (Jaworski et

al., 2015; Pak et al., 2020; Zelina et al., 2014). Another study used binding ELISA to confirm that in murine BMDM, ROBO1, but not ROBO3, binds to NSlit2 (Park et al., 2019). We have now clarified these points in the Introduction section of our paper (please see page 4 of the revised manuscript).

Having established that ROBO1 is the only binding partner of NSlit2 in murine macrophages, we next sought to evaluate the role of ROBO1 in NSlit2-induced effects on macrophages. We found that knockdown of Robo1 in RAW264.7 murine macrophages, using a specific siRNA, resulted in loss of Slit2's actions on macrophage spreading and rounding (Fig. 1G-H). We then confirmed these findings using ROBO1N, a soluble N-terminal fragment of ROBO1 which contains the Slit-binding IG1 domain (Morlot et al., 2007), and which has been previously shown to block the binding of NSlit2 to Robo1 *in vitro* (Patel et al., 2012; Werbowetski-Ogilvie et al., 2006). In the present study, we found that incubation of NSlit2 with ROBO1N also prevented NSlit2's effects on macrophage spreading and rounding (Supplementary Fig. 1F-G). To further verify our findings, we used recombinant Slit2ΔD2 protein as a negative control in all *in vitro* and *in vivo* experiments. As described in the schematic of Fig. 1B, Slit2ΔD2 lacks the Robo-binding D2 domain (Chedotal, 2007) and it has been replaced by a linker. We and others have previously shown that Slit2ΔD2 does not bind ROBO1 (Anand et al., 2013; Patel et al., 2012).

To further validate the role of ROBO1 in SLIT2's actions on macropinocytosis, we used two KRAS-transformed cancer cell lines. NSlit2 inhibited macropinocytosis in ROBO1-expressing PANC1 cells (Gohrig et al., 2014), but not in ROBO1-deficient DLD1 cells (Zhou et al., 2011).

Collectively, these results indicate that the effects of Slit2 on macrophage spreading and rounding, and on cell macropinocytosis, occur through ROBO1. We have now clarified these important points. Please see Fig. 6B-D and page 14 of the revised manuscript.

According to the methods, Slit2 proteins were injected at 30 nM but the authors do not say anything about the injected volume. How was the concentration (dose) determined and what is the half-life of the 3 proteins injected.

We apologize for the confusing terminology. We prepared equimolar concentrations of all recombinant Slit proteins (30 nM) in 1 ml of saline. The final volume injected per mouse per injection was 200 μl with 1 μg NSlit2, 1 μg Slit2ΔD2, and 0.2 μg CSlit2. This has been corrected in the Methods section. Please see 'Methods' and Fig. 4-5 legends of the revised manuscript.

The half-life of injected Slit proteins in the body is not known. However, our group has previously shown that the effect of exogenously administered recombinant NSlit2 on monocyte/macrophage migration can last for at least 4 days (Mukovozov et al., 2015). Of note, in our current study, the effects of injected NSlit2 were assessed in less than 24 hours, well within the previously documented period of activity.

Slit2 should also act on vessel permeability.

All *in vivo* protocols in our study involved a single injection of 1μg of recombinant NSlit2.

The effect of an injection of recombinant NSlit2 on vascular permeability has not been fully elucidated. Nonetheless, London *et al.* have reported that NSlit2 stabilized endothelial cell-cell junctions *in vitro* by inhibiting pro-inflammatory cytokine-induced phosphorylation of VE-cadherin (London et al., 2010). Moreover, when administered *in vivo* in the presence of inflammatory stimuli, NSlit2 prevented an inflammation-induced increase in vascular permeability. In these *in vivo* studies, the authors used doses of recombinant NSlit2 similar to those we used in our current study (please see summary table below). The authors concluded that the actions of SLIT2 on vascular stabilization were mediated by the endothelial-cell-specific Roundabout receptor, ROBO4 (London et al., 2010).

In vivo Inflammation Model (London et al., 2010)	Dose of NSlit2 per injection	Frequency	Biological Effect
1. LPS-induced lung injury (LPS in saline solution was administered intra-tracheally)	3.5 µg	single injection i.v.	Fig. 3A – Decrease in LPS- induced vascular permeability 5 h after the LPS dose
2. CLP (cecal ligation & puncture) -induced polymicrobial sepsis	5 µg	single injection i.p.	Fig. 4A – Decrease in CLP- induced vascular permeability 4 h after the CLP injury
3. H5N1 infection model	1.56 µg	Daily i.v.	Fig. 5A – Decrease in H5N1- induced vascular permeability 3 days after H5N1 infection

However, in the study by London *et al.*, the effects of NSlit2 alone, in the absence of inflammatory injury, were not investigated. The observed actions on the vasculature were proposed to occur through the endothelial cell-specific Roundabout receptor, ROBO4 (Jones et al., 2009), but a more recent study challenges this idea (Zhang et al., 2016). Overall, the role of SLIT2 in maintaining vascular integrity under basal conditions is not clear. Furthermore, the relative contributions of endothelial ROBO1 and ROBO2 in mediating SLIT2's different actions on the vasculature also remain to be elucidated (Rama et al., 2015). We have now clarified these important points in the revised manuscript (please see pages 18-19 of the revised manuscript).

The authors also need to show the CD68 staining together with the dextran.

We thank the reviewer for this suggestion and have now included these images in Supplementary Fig. 4G.

The authors say that Robo1 KO mice are not viable, but this is not the case and several viable lines exist. See for instance J Clin Invest. 2018;128(4):1429-1441 where studies were performed in Robo1^{-/-} adult mice. Likewise, Slit2 conditional lines also exist and were used by many groups (see for example Pinho et al, Nat Commun. 2018 Nov 30;9(1):5083).

Pinho et al. (Nat Commun. 2018 Nov 30;9(1):5083) used Slit1^{-/-} and not Slit2^{-/-} mice in their study. The authors note that “Since Slit2 knockout animals are not viable, we proceeded studying Slit1 knockout (Slit1^{-/-}) mice” (Pinho et al., 2018). In keeping with this, Tessier-Lavigne *et al.* have previously shown that global Slit2^{-/-} knockout mice do not survive beyond 2 weeks (Plump et al., 2002). The predominant *in vivo* source of Slit2 remains a topic of active investigation. A recent paper suggests that endothelial cells are the predominant source of both SLIT2 and SLIT3 proteins in the kidney (Wang et al., 2020). However, conditional Slit2 knockout mice, with no developmental abnormalities, are currently not available.

Recently, a Slit2 allele, Slit2^{em1(IMPC)Mbp}, with endonuclease (Cas9)-mediated knockout has been generated at the Mouse Biology Program at UC Davis. The heterozygous mice, with one normal copy of Slit2, are born without any anomaly. However, characteristics of homozygous adult mice carrying this mutation have not yet been described in the peer-reviewed scientific literature.

Slit2^{em1(IMPC)Mbp} allele information - <http://www.informatics.jax.org/allele/MGI:6316121>

In the study mentioned by the reviewer, Kim BJ et al. used genetically-altered mice deficient in Robo1 (*J Clin Invest.* 2018;128(4):1429-1441) (Kim et al., 2018). However, these Robo1^{-/-} mice are not a complete knockout but a hypomorphic, partial deletion, originally developed by Dr. Tessier-Lavigne's group (Long et al., 2004). Adult mice with the hypomorphic deletion of Robo1, have osteopenia due to decreased bone formation and increased bone resorption (Kim et al., 2018). The same group has recently reported that these effects on bone density might be jointly mediated by SLIT2 and SLIT3 acting via ROBO1 (Park et al., 2019). However, the individual contributions of the two SLIT proteins in bone formation *in vivo* remain to be elucidated.

Several other groups have reported that mice completely lacking ROBO1 are not viable in the short or long term. One such study demonstrated that Robo1^{-/-} mice die in the perinatal period due to inadequate lung development (Xian et al., 2001). In addition, a group led by Dr. Linda Richards reported that homozygous Robo1^{-/-} mice created using Cre-loxP recombination system do not reach adulthood (Andrews et al., 2006). Myeloid cell-specific Robo1 knockout are currently not available.

The study mentioned by the reviewer (Pinho et al, *Nat Commun.* 2018 Nov 30;9(1):5083) investigated the function of SLIT1 in pancreatic cancer. However, cellular sources of endogenous SLIT1 in peripheral organs remain unknown. Furthermore, as mentioned earlier, due to the lack of validated reagents to measure SLIT1 protein levels, recent studies have only been able to report its mRNA levels (Pinho et al., 2018; Shuai et al., 2018). When such reagents become available, the role of SLIT1 in peripheral organs can be investigated. We have clarified this important point on page 19 of the revised manuscript.

Overall, genetically modified mice completely lacking SLIT2 or ROBO1 are not available, and could, therefore, not be used for our studies. We have now clarified these important points in the revised version of the manuscript (please see page 14 in the revised manuscript).

Reviewer #3 (Remarks to the Author):

This manuscript is considerably improved compared with the previous version. The data are better presented, more comprehensive and it is much easier to read. There is now at least some evidence that Slit2/Robo signalling regulates macropinocytosis and downstream effects in vivo. The experiments where Slit ligands are injected into mice are really just using the mouse as a 'test tube' and are less convincing than those where levels and effects of endogenous Slit ligand are probed. The data here are quite limited and found in Figure 5H but at least they seem to show that endogenous Slit has real effects on MDP signalling and it is plausible that this is mediated by effects on endogenous macropinocytosis of the ligand. For me this was an important new experiment.

We thank the reviewer for these positive comments. We really appreciated the reviewer's suggestion to examine the effects of endogenous SLIT2 on macropinocytosis, and agree that the new findings significantly clarify how native SLIT2 regulates macropinocytosis.

On a minor point of clarification, the authors refer a couple of times to the stimulation of macropinocytosis by pathogen-derived ligands like LPS (e.g on page 3). This is only part of the story as anyone familiar with the dendritic cell field will tell you. The key early experiments of Lanzavecchia and colleagues using human DC showed that LPS down-regulates macropinocytosis (Sallusto et al *JExpMed*, 1995) over the long term (a finding confirmed by others) and it was only subsequently shown (by West et al *Science*, 2004) that prior to this downregulation, there is a transient stimulation of macropinocytosis by TLR

ligands. This was why I previously asked whether Slit/Robo signalling might be involved in long-term suppression of macropinocytosis in dendritic cells.

We thank the reviewer for raising this important point. On page 3, in the 'Introduction' section we have now replaced mention of LPS with that of IpaC from Shigella, known to induce macropinocytosis (Mattock and Blocker, 2017).

We have also now described transient and sustained effects of TLR ligands (LPS) on macropinocytosis in the 'Discussion' section. Please see page 18 of the revised manuscript. Guan et al. reported that SLIT2-ROBO1 signaling inhibits dendritic cell migration *in vitro* and *in vivo*, and Prasad et al. demonstrated that SLIT2 inhibits podosome formation in immature dendritic cells by sequestering proteins away from WASP-ARP2/3-Actin complex (Guan et al., 2003; Prasad et al., 2012). Notably, endogenous *Slit2* mRNA is up-regulated following antigen sensitization *in vivo* but the physiological significance of this phenomenon remains to be elucidated (Guan et al., 2003; Ye et al., 2010). In the present study, we focused on intracellular NOD2 signaling because MDP, a NOD2-specific ligand, is known to enter primary macrophages predominantly via macropinocytosis (Canton et al., 2016). Although beyond the scope of the current study, in future studies, we will investigate the role of SLIT2 on transient versus sustained suppression of macropinocytosis in dendritic cells, and its impact on antigen uptake. We have addressed this important point in the revised manuscript (please see page 18 of the revised manuscript).

Overall I felt this study was worthy of publication.

We thank the reviewer for the positive comment about the revised manuscript.

There is a typo on line 273 which suggests spell checking/proof reading is needed.

We apologize for the typographical errors. We have now corrected this error (please see pages-12-13 of the revised manuscript), and more carefully proof read the entire manuscript to correct any other typographical errors.

We would like to thank all reviewers for allowing us to address the concerns noted. Based on the previous comments from each expert reviewer, we have again performed additional experiments that clarify our original findings. Thank you for allowing us to again submit our revised manuscript for your consideration. We hope that you will favourably review our work and now find it suitable for publication in Nature Communications.

References:

- Anand, A.R., Zhao, H., Nagaraja, T., Robinson, L.A., and Ganju, R.K. (2013). N-terminal Slit2 inhibits HIV-1 replication by regulating the actin cytoskeleton. *Retrovirology* 10, 2.
- Andrews, W., Liapi, A., Plachez, C., Camurri, L., Zhang, J., Mori, S., Murakami, F., Parnavelas, J.G., Sundaresan, V., and Richards, L.J. (2006). Robo1 regulates the development of major axon tracts and interneuron migration in the forebrain. *Development* 133, 2243-2252.
- Burgstaller, G., Oehrle, B., Gerckens, M., White, E.S., Schiller, H.B., and Eickelberg, O. (2017). The instructive extracellular matrix of the lung: basic composition and alterations in chronic lung disease. *Eur Respir J* 50.
- Canton, J., Schlam, D., Breuer, C., Gutschow, M., Glogauer, M., and Grinstein, S. (2016). Calcium-sensing receptors signal constitutive macropinocytosis and facilitate the uptake of NOD2 ligands in macrophages. *Nat Commun* 7, 11284.
- Chang, J., Lan, T., Li, C., Ji, X., Zheng, L., Gou, H., Ou, Y., Wu, T., Qi, C., Zhang, Q., *et al.* (2015). Activation of Slit2-Robo1 signaling promotes liver fibrosis. *J Hepatol* 63, 1413-1420.
- Chaturvedi, S., Yuen, D.A., Bajwa, A., Huang, Y.W., Sokollik, C., Huang, L., Lam, G.Y., Tole, S., Liu, G.Y., Pan, J., *et al.* (2013). Slit2 prevents neutrophil recruitment and renal ischemia-reperfusion injury. *J Am Soc Nephrol* 24, 1274-1287.
- Chedotal, A. (2007). Slits and their receptors. *Adv Exp Med Biol* 621, 65-80.
- Geutskens, S.B., Hordijk, P.L., and van Hennik, P.B. (2010). The chemorepellent Slit3 promotes monocyte migration. *J Immunol* 185, 7691-7698.
- Gohrig, A., Detjen, K.M., Hilfenhaus, G., Korner, J.L., Welzel, M., Arsenic, R., Schmuck, R., Bahra, M., Wu, J.Y., Wiedenmann, B., *et al.* (2014). Axon guidance factor SLIT2 inhibits neural invasion and metastasis in pancreatic cancer. *Cancer Res* 74, 1529-1540.
- Guan, H., Zu, G., Xie, Y., Tang, H., Johnson, M., Xu, X., Kevil, C., Xiong, W.C., Elmets, C., Rao, Y., *et al.* (2003). Neuronal repellent Slit2 inhibits dendritic cell migration and the development of immune responses. *J Immunol* 171, 6519-6526.
- Jaworski, A., Tom, I., Tong, R.K., Gildea, H.K., Koch, A.W., Gonzalez, L.C., and Tessier-Lavigne, M. (2015). Operational redundancy in axon guidance through the multifunctional receptor Robo3 and its ligand NELL2. *Science* 350, 961-965.
- Jones, C.A., Nishiya, N., London, N.R., Zhu, W., Sorensen, L.K., Chan, A.C., Lim, C.J., Chen, H., Zhang, Q., Schultz, P.G., *et al.* (2009). Slit2-Robo4 signalling promotes vascular stability by blocking Arf6 activity. *Nat Cell Biol* 11, 1325-1331.
- Kanellis, J., Garcia, G.E., Li, P., Parra, G., Wilson, C.B., Rao, Y., Han, S., Smith, C.W., Johnson, R.J., Wu, J.Y., *et al.* (2004). Modulation of inflammation by slit protein in vivo in experimental crescentic glomerulonephritis. *Am J Pathol* 165, 341-352.
- Kim, B.J., Lee, Y.S., Lee, S.Y., Baek, W.Y., Choi, Y.J., Moon, S.A., Lee, S.H., Kim, J.E., Chang, E.J., Kim, E.Y., *et al.* (2018). Osteoclast-secreted SLIT3 coordinates bone resorption and formation. *J Clin Invest* 128, 1429-1441.

- London, N.R., Zhu, W., Bozza, F.A., Smith, M.C., Greif, D.M., Sorensen, L.K., Chen, L., Kaminoh, Y., Chan, A.C., Passi, S.F., *et al.* (2010). Targeting Robo4-dependent Slit signaling to survive the cytokine storm in sepsis and influenza. *Sci Transl Med* 2, 23ra19.
- Long, H., Sabatier, C., Ma, L., Plump, A., Yuan, W., Ornitz, D.M., Tamada, A., Murakami, F., Goodman, C.S., and Tessier-Lavigne, M. (2004). Conserved roles for Slit and Robo proteins in midline commissural axon guidance. *Neuron* 42, 213-223.
- Mattock, E., and Blocker, A.J. (2017). How Do the Virulence Factors of Shigella Work Together to Cause Disease? *Front Cell Infect Microbiol* 7, 64.
- Morlot, C., Thielens, N.M., Ravelli, R.B., Hemrika, W., Romijn, R.A., Gros, P., Cusack, S., and McCarthy, A.A. (2007). Structural insights into the Slit-Robo complex. *Proc Natl Acad Sci U S A* 104, 14923-14928.
- Mukovozov, I., Huang, Y.W., Zhang, Q., Liu, G.Y., Siu, A., Sokolsky, Y., Patel, S., Hyduk, S.J., Kutryk, M.J., Cybulsky, M.I., *et al.* (2015). The Neurorepellent Slit2 Inhibits Postadhesion Stabilization of Monocytes Tethered to Vascular Endothelial Cells. *J Immunol* 195, 3334-3344.
- Pak, J.S., DeLoughery, Z.J., Wang, J., Acharya, N., Park, Y., Jaworski, A., and Ozkan, E. (2020). NELL2-Robo3 complex structure reveals mechanisms of receptor activation for axon guidance. *Nat Commun* 11, 1489.
- Park, S.J., Lee, J.Y., Lee, S.H., Koh, J.M., and Kim, B.J. (2019). SLIT2 inhibits osteoclastogenesis and bone resorption by suppression of Cdc42 activity. *Biochem Biophys Res Commun* 514, 868-874.
- Patel, S., Huang, Y.W., Rehem, A., Pluthero, F.G., Chaturvedi, S., Mukovozov, I.M., Tole, S., Liu, G.Y., Li, L., Durocher, Y., *et al.* (2012). The cell motility modulator Slit2 is a potent inhibitor of platelet function. *Circulation* 126, 1385-1395.
- Pertz, O., Hodgson, L., Klemke, R.L., and Hahn, K.M. (2006). Spatiotemporal dynamics of RhoA activity in migrating cells. *Nature* 440, 1069-1072.
- Pinho, A.V., Van Bulck, M., Chantrill, L., Arshi, M., Sklyarova, T., Herrmann, D., Vennin, C., Gallego-Ortega, D., Mawson, A., Giry-Laterriere, M., *et al.* (2018). ROBO2 is a stroma suppressor gene in the pancreas and acts via TGF-beta signalling. *Nat Commun* 9, 5083.
- Plump, A.S., Erskine, L., Sabatier, C., Brose, K., Epstein, C.J., Goodman, C.S., Mason, C.A., and Tessier-Lavigne, M. (2002). Slit1 and Slit2 cooperate to prevent premature midline crossing of retinal axons in the mouse visual system. *Neuron* 33, 219-232.
- Prasad, A., Kuzontkoski, P.M., Shrivastava, A., Zhu, W., Li, D.Y., and Gropman, J.E. (2012). Slit2N/Robo1 inhibit HIV-gp120-induced migration and podosome formation in immature dendritic cells by sequestering LSP1 and WASp. *PLoS One* 7, e48854.
- Rama, N., Dubrac, A., Mathivet, T., Ni Charthaigh, R.A., Genet, G., Cristofaro, B., Pibouin-Fragner, L., Ma, L., Eichmann, A., and Chedotal, A. (2015). Slit2 signaling through Robo1 and Robo2 is required for retinal neovascularization. *Nat Med* 21, 483-491.
- Shuai, W., Wu, J., Chen, S., Liu, R., Ye, Z., Kuang, C., Fu, X., Wang, G., Li, Y., Peng, Q., *et al.* (2018). SUV39H2 promotes colorectal cancer proliferation and metastasis via tri-methylation of the SLIT1 promoter. *Cancer Lett* 422, 56-69.
- Swanson, J.A. (2008). Shaping cups into phagosomes and macropinosomes. *Nat Rev Mol Cell Biol* 9, 639-649.

Tole, S., Mukovozov, I.M., Huang, Y.W., Magalhaes, M.A., Yan, M., Crow, M.R., Liu, G.Y., Sun, C.X., Durocher, Y., Glogauer, M., *et al.* (2009). The axonal repellent, Slit2, inhibits directional migration of circulating neutrophils. *J Leukoc Biol* 86, 1403-1415.

van Helden, S.F., Anthony, E.C., Dee, R., and Hordijk, P.L. (2012). Rho GTPase expression in human myeloid cells. *PLoS One* 7, e42563.

Wang, X., Liu, J., Yin, W., Abdi, F., Pang, P.D., Fucci, Q.A., Abbott, M., Chang, S.L., Steele, G., Patel, A., *et al.* (2020). miR-218 Expressed in Endothelial Progenitor Cells Contributes to the Development and Repair of the Kidney Microvasculature. *Am J Pathol*.

Werbowski-Ogilvie, T.E., Seyed Sadr, M., Jabado, N., Angers-Loustau, A., Agar, N.Y., Wu, J., Bjerkvig, R., Antel, J.P., Faury, D., Rao, Y., *et al.* (2006). Inhibition of medulloblastoma cell invasion by Slit. *Oncogene* 25, 5103-5112.

Wu, J.Y., Feng, L., Park, H.T., Havlioglu, N., Wen, L., Tang, H., Bacon, K.B., Jiang, Z., Zhang, X., and Rao, Y. (2001). The neuronal repellent Slit inhibits leukocyte chemotaxis induced by chemotactic factors. *Nature* 410, 948-952.

Xian, J., Clark, K.J., Fordham, R., Pannell, R., Rabbitts, T.H., and Rabbitts, P.H. (2001). Inadequate lung development and bronchial hyperplasia in mice with a targeted deletion in the Dutt1/Robo1 gene. *Proc Natl Acad Sci U S A* 98, 15062-15066.

Xu, R., Yallowitz, A., Qin, A., Wu, Z., Shin, D.Y., Kim, J.M., Debnath, S., Ji, G., Bostrom, M.P., Yang, X., *et al.* (2018). Targeting skeletal endothelium to ameliorate bone loss. *Nat Med* 24, 823-833.

Yao, Y., Zhou, Z., Li, L., Li, J., Huang, L., Li, J., Qi, C., Zheng, L., Wang, L., and Zhang, Q.Q. (2019). Activation of Slit2/Robo1 Signaling Promotes Tumor Metastasis in Colorectal Carcinoma through Activation of the TGF-beta/Smads Pathway. *Cells* 8.

Ye, B.Q., Geng, Z.H., Ma, L., and Geng, J.G. (2010). Slit2 regulates attractive eosinophil and repulsive neutrophil chemotaxis through differential srGAP1 expression during lung inflammation. *J Immunol* 185, 6294-6305.

Zelina, P., Blockus, H., Zagar, Y., Peres, A., Friocourt, F., Wu, Z., Rama, N., Fouquet, C., Hohenester, E., Tessier-Lavigne, M., *et al.* (2014). Signaling switch of the axon guidance receptor Robo3 during vertebrate evolution. *Neuron* 84, 1258-1272.

Zeng, Z., Wu, Y., Cao, Y., Yuan, Z., Zhang, Y., Zhang, D.Y., Hasegawa, D., Friedman, S.L., and Guo, J. (2018). Slit2-Robo2 signaling modulates the fibrogenic activity and migration of hepatic stellate cells. *Life Sci* 203, 39-47.

Zhang, F., Prahst, C., Mathivet, T., Pibouin-Fragner, L., Zhang, J., Genet, G., Tong, R., Dubrac, A., and Eichmann, A. (2016). The Robo4 cytoplasmic domain is dispensable for vascular permeability and neovascularization. *Nat Commun* 7, 13517.

Zhou, W.J., Geng, Z.H., Chi, S., Zhang, W., Niu, X.F., Lan, S.J., Ma, L., Yang, X., Wang, L.J., Ding, Y.Q., *et al.* (2011). Slit-Robo signaling induces malignant transformation through Hakai-mediated E-cadherin degradation during colorectal epithelial cell carcinogenesis. *Cell Res* 21, 609-626.

REVIEWERS' COMMENTS

Reviewer #1 (Remarks to the Author):

The authors have addressed all my comments.

Reviewer #2 (Remarks to the Author):

Some pieces of the puzzle are still missing, but I think the authors did their best to address my concerns

Reviewer #1 (Remarks to the Author):

The authors have addressed all my comments.

We thank the reviewer and are pleased that the revised manuscript has adequately addressed all of the concerns noted.

Reviewer #2 (Remarks to the Author):

Some pieces of the puzzle are still missing, but I think the authors did their best to address my concerns.

We thank the reviewer for noting our sincere efforts to address each of the concerns, and for now finding our work suitable for publication in *Nature Communications*.